# Period-doubling in the phase dynamics of a shunted HgTe quantum well Josephson junction

Wei Liu [1,2], Stanislau U. Piatrusha [1,2], Xianhu Liang[1,2], Sandeep Upadhyay[1,2], Lena Fürst[1,2], Charles Gould [1,2], Johannes Kleinlein [1,2], Hartmut Buhmann [1,2], Martin P. Stehno [1,2] ✉ & Laurens W. Molenkamp [1,2]

The fractional AC Josephson effect is a discerning property of topological superconductivity in hybrid Josephson junctions. Recent experimental observations of missing odd Shapiro steps and half Josephson frequency emission in various materials have sparked significant debate regarding their potential origin in the effect. In this study, we present microwave emission measurements on a resistively shunted Josephson junction based on a HgTe quantum well. We demonstrate that, with significant spurious inductance in the shunt wiring, the experiment operates in a nonlinear dynamic regime characterized by period-doubling. This leads to additional microwave emission peaks at half of the Josephson frequency, $f_J/2$, which can mimic the $4\pi$-periodicity of topological Andreev states. The observed current-voltage characteristics and emission spectra are well-described by a simple RCLSJ model. Furthermore, we show that the nonlinear dynamics of the junction can be controlled using gate voltage, magnetic field, and temperature, with our model accurately reproducing these effects without incorporating any topological attributes. Our observations urge caution in interpreting emission at $f_J/2$ as evidence for gapless Andreev bound states in topological junctions and suggest the appropriate parameter range for future experiments.

The report of a fractional AC Josephson effect in Nb/InSn nanowire/Nb devices by Rokhinson, Liu, and Furdyna[1] has directed attention to transport measurements on RF-driven Josephson junctions as a means of detecting topological states in candidate systems for Majorana physics. Overlapping Majoranas form mid-gap Andreev bound states (ABS) that transport single electrons across the junction. This results in super-current transport with $4\pi$-phase-periodicity that can be detected in the phase dynamics of the RF-driven device[2]. The phase evolution locks to twice the fundamental Josephson frequency, leading to Shapiro steps with spacing $hf/e$ for drive frequency $f$. Subsequently, the suppression of Shapiro steps for odd multiples of $hf/2e$ has been observed in Josephson junctions using a wide variety of topological materials as weak link,

including work on HgTe Josephson devices in our group[3,4]. The results are unexpected as time-reversal symmetry (TRS) is not broken in these experiments (cf. ref. [5] and later work), and the Fermi level resides in the conduction band when the signal is strongest.

TRS-breaking by a large magnetic field localized in the topological insulator opens a gap in the ABS dispersion[5] and prevents pumping of quasiparticles across the superconducting gap when finite bias is applied[6]. Alternative methods for preventing quasiparticle pumping involve inelastic scattering between the Andreev levels[6] and dynamic effects[7] that restore the $4\pi$-periodicity.

Conversely, there exist mechanisms that lead to a $4\pi$-periodic supercurrent in trivial Josephson devices, e.g., Landau-Zener

[1]Physikalisches Institut (EP3), Universität Würzburg, Am Hubland, 97074 Würzburg, Germany. [2]Institute for Topological Insulators, Am Hubland, 97074 Würzburg, Germany. ✉e-mail: martin.stehno@uni-wuerzburg.de

transitions between Andreev levels may give rise to a $4\pi$-periodicity[2,8–10], especially in semiconductor Josephson junctions with highly transparent interfaces. Missing Shapiro steps in topologically trivial InAs quantum well Josephson devices have been attributed to this effect[11]. Moreover, nonlinearities in the bias-dependent resistance may also affect the visibility of Shapiro steps[12]. On top of these uncertainties, some topological junctions do not show $4\pi$-periodicity[13].

An alternative, more direct method for studying the AC Josephson effect is the detection of microwave photons emitted by the voltage-biased Josephson junction[14,15]. Experiments on topological HgTe quantum well (QW) Josephson junctions detect emission at half of the Josephson frequency ($f_J/2$)[16], consistent with the earlier reports of $4\pi$-phase-periodicity in Shapiro step measurements[4]. A $4\pi$-periodic Josephson effect in photon emission has also been reported for InAs nanowire junctions[17], where an on-chip detector was used. The seemingly conflicting aspects of earlier theoretical and experimental results, and the more recent experiments on other material systems motivate our renewed interest in the topic, extending the scope of our analysis to nonlinear dynamics effects.

Driven Josephson junctions exhibit a wide range of nonlinear dynamics phenomena, including period-doubling sequences, relaxation oscillations, metastable states, and chaos[18–24]. This has made Josephson junctions a model tool for theoretical and experimental research on nonlinear systems. In many cases of interest, the complex dynamics must not be understood as an intrinsic property of the junction, but it arises from embedding the device in an external biasing circuit or electrodynamic environment. The importance of the environment has been recognized early in the description of the dynamics of small Josephson tunneling junctions. In this case, the charging energy competes with the Josephson energy[25,26]. Recently, this approach has been used to demonstrate AC-driven current steps ("dual Shapiro steps") in the current-voltage characteristic ($I$–$V$) of Josephson device circuits by carefully engineering the junction

environment, thus opening a promising route towards a quantum current standard[27–30]. Coupling voltage-biased Josephson junctions to microwave resonators can lead to nonlinear effects such as multi-photon emission[31,32]. With additional microwave driving, nonlinear dynamics due to strong coupling between photons and the nonlinear system holds great potential for realizing quantum-limited amplification[33] and AC Josephson junction lasing[34].

In this article, we report extensive measurements of microwave emissions from a new generation of HgTe QW Josephson junctions in a newly-built RF measurement setup in our group. The experiments are carried out by shunting the junction with a commercial thin-film resistor in two wiring configurations that differ by the amount of spurious wiring inductance. The wiring layout with large parallel inductance exhibits microwave emission at $f_J/2$ in addition to a signal at $f_J$. We show that the current-voltage characteristics and emission spectra closely match a model based on a resistively-, capacitively-, and inductively-shunted Josephson junction (RCLSJ) with a $\sin\varphi$ current-phase relationship. We present numerical calculations that suggest the system is close to a period-doubling instability, causing the $4\pi$-periodic evolution of the junction phase $\varphi$. We test this hypothesis experimentally by tuning the junction parameters with a gate, magnetic field, or temperature. This allows us to cross over into a regime with lowest frequency emission at $f_J$, indicating a transition to period-one phase evolution. In a complimentary experiment, we glue the shunting resistor on-chip, thus significantly reducing the circuit inductance. In this configuration, the emission spectra exhibit a pronounced peak at $f_J$, while emission at $f_J/2$ is absent. Our analysis demonstrates that comprehensive circuit modeling is of great relevance for the interpretation of microwave experiments on topological Josephson junctions.

## Results

We study a side-contacted HgTe QW Josephson device (Fig. 1a, b). The DC transport characterization is performed in a dilution refrigerator

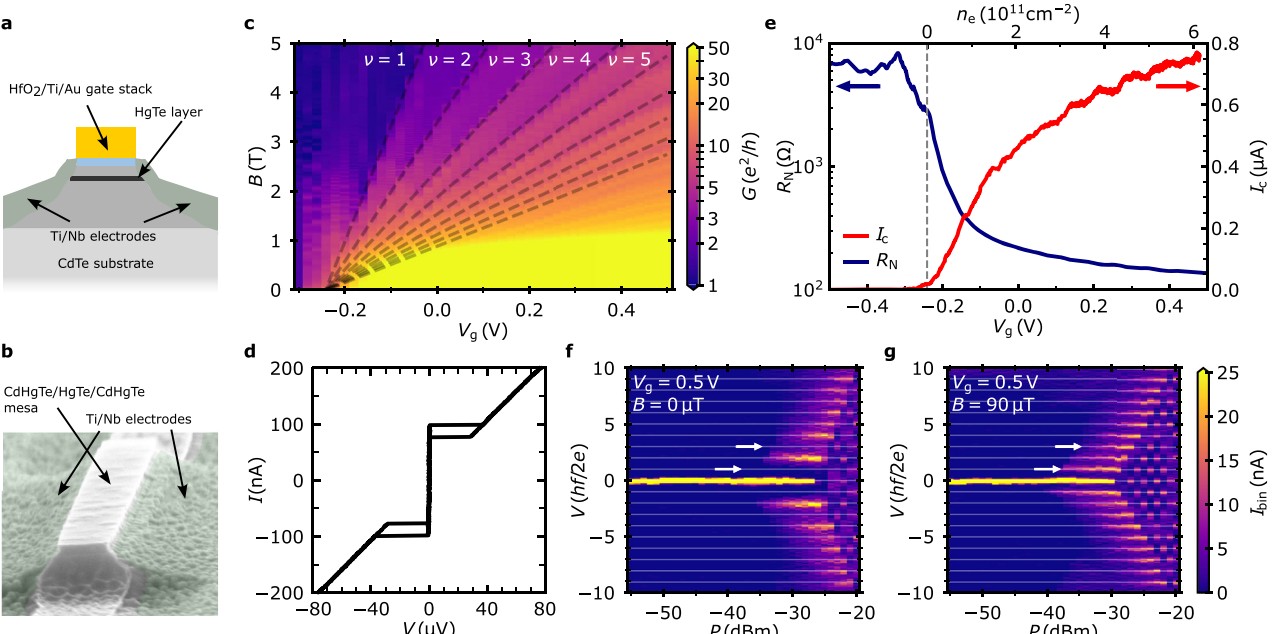

**Fig. 1 | Sample and DC characterization. a** Schematic of the side-contacted HgTe QW Josephson junction; **b** false-colored SEM micrograph of a side-contacted junction with an electrode separation of ~280 nm but without gate; **c** 2D map of the junction differential conductance $G$ as a function of gate voltage $V_g$ and magnetic field $B$; conductance steps are labeled by the Landau level index $\nu$; **d** hysteretic I-V of the unshunted junction [Fermi level identical to Fig. 3b for the shunted configuration]; **e** gate dependence of the normal state resistance $R_N$ and critical current $I_c$ as a function of gate voltage $V_g$ [or bulk carrier density $n_e$]; **f** 2D map of the voltage histogram of Shapiro steps (bin size: 0.33 μV) as a function of microwave power $P$ and voltage $V$ for frequency $f = 1.6$ GHz at $V_g = 0.5$ V; first and third Shapiro steps are missing (positions indicated by arrows); **g** same as **f** with perpendicular magnetic field $B = 90$ μT.

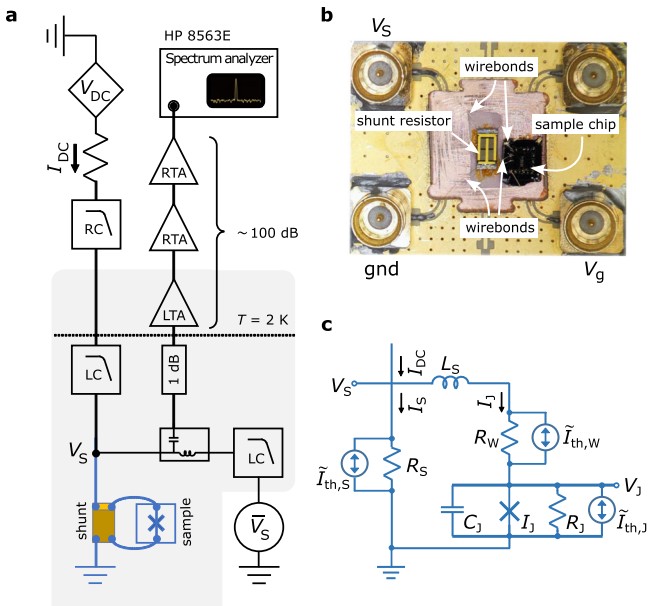

**Fig. 2 | RF measurement circuit (C1). a** Schematic of the RF measurement circuit [gate connection not shown]. The bias current $I_{DC}$ is generated by applying a voltage $V_{DC}$ via a series resistor, followed by RC- and LC- low-pass filters (RC, LC). The voltage across the shunted sample $V_S$ is amplified using an amplifier chain with a low-temperature (LTA), anchored at temperature $T = 2$ K, and two room-temperature amplifiers (RTA) with a total gain of ~ 100 dB. An attenuator (1 dB) provides thermal anchoring. The DC-averaged component $\overline{V}_S$ is measured separately. **b** Photo of sample and shunt resistor mounted on and wired to the RF circuit board; **c** equivalent circuit used in numerical simulations.

with heavily-filtered measurement leads at the base temperature, $T = 35$ mK. The self-aligned, side-contacted device fabrication technique allows us to gate the weak link reliably into the bulk gap of the quantum spin Hall (QSH) material. To demonstrate this, we map out the conductance $G$ of the Josephson junction as a function of gate voltage $V_g$ in a perpendicular magnetic field $B$ (Fig. 1c). A clear sequence of conductance steps is observed as Landau levels are depopulated with increasing $B$. The Landau level fan extrapolates to the charge neutrality point at $V_g \approx -0.245$ V where the Fermi level reaches the bottom of the first conduction band subband.

At $B = 0$, the current-biased device exhibits the hysteretic current-voltage ($I–V$) characteristic (Fig. 1d) of an underdamped Josephson junction[35,36], cf. Section S1 of the Supplementary Information (SI). The gate dependence of the critical current $I_c$ and the normal state resistance $R_N$ are depicted in Fig. 1e. Here, $I_c$ is defined by the voltage criterion $|V| < 1.5\,\mu$V, and $R_N$ is the slope of the $I–V$ in the linear region, $V \gtrsim 0.4$ mV. In the band gap, the magnitude of the critical current is of the order of a few nA and fluctuates with $V_g$. It increases steeply as the first subband is populated with carriers. Concurrently, $R_N$ decreases by almost two orders of magnitude as the gate voltage is increased. We study Shapiro steps by irradiating the sample with microwaves of frequency $f$. Figure 1f depicts a color plot of the current histogram of the $I–V$ as function of microwave power $P$ and junction voltage $V$ at $V_g = 0.5$ V, i.e., with the Fermi level high in the conduction band. It features voltage steps at $n \times hf/2e$, for integer values $n$, where $h$ and $e$ denote Planck's constant and the electronic charge. Notably, the first ($n = \pm 1$) and third ($n = \pm 3$) steps are missing [cf. Section S1 of the SI]. The conventional pattern of Shapiro steps is recovered by applying a small magnetic field $B = 90\,\mu$T (Fig. 1g). This reproduces the observations in refs. 3,4.

Next we perform measurements of the microwave emission of the junction. A schematic of the emission measurement is shown in Fig. 2a.

So as to provide stable voltage biasing to a device in conventional four-terminal leads geometry, we connect a commercial surface-mount resistor in parallel. Measurement circuit C1 is realized by gluing the device and a thin-film shunting resistor side-by-side onto a RF-circuit board (Fig. 2b). Electrical connections are made by placing wirebonds. The AC and DC components of the sample voltage $V_S$ are measured by separate measurement circuits connected to the shunted sample via a bias tee. Due to standing wave conditions in the wiring, the effective RF gain rapidly oscillates with frequency. Thus we plot the power spectral density of the RF signal normalized by the maximum amplitude after subtracting the noise background for each detection frequency in all 2D maps (cf. Section S3 of the SI for details on background subtraction and data normalization).

A simple lumped-element circuit model of the externally-shunted Josephson junction (circuit C1) is depicted in Fig. 2c. Here, quantities with indices J, W, and S refer to the junction, on-chip wiring layer including bond pads, and external shunt, respectively. Node voltages and branch currents in Fig. 2c are fluctuating quantities. Below we introduce barred quantities to refer to the DC averaged component of a fluctuating quantity, where necessary.

The wirebonds add an inductance $L_S$ and the on-chip wiring layer a resistance $R_W$ in series with the junction. The Josephson junction is represented by the phase-dependent supercurrent $I_X$, the resistance $R_J$ that takes into account the quasiparticle current of the junction, and the geometric capacitance $C_J$ of the device, all connected in parallel. The white noise current sources $\tilde{I}_{th,x}$ in parallel to resistors $R_x$, $x = \{J, W, S\}$, are introduced for modeling thermal fluctuations in the numerical simulations below.

First, the resistance values of C1 (Fig. 2c) are determined in $I–V$ measurements. The shunting resistance $R_S = 8.27\,\Omega$ is obtained upon gating the junction into the bulk gap, where $I_X \approx 0$, $R_J \sim h/2e^2$, and the current in the junction branch $I_J$ is approximately zero. The parameters for the wiring resistance $R_W = 10.23\,\Omega$ and the (small bias) subgap junction resistance $R_J(V_g)$ are found by extracting the slopes of the measured $I_{DC}(\overline{V}_S)$ curves at $I_{DC} = 0$ and $\overline{V}_J \gtrsim 40\,\mu$V, respectively. Here, $\overline{V}_J = \overline{V}_S - \overline{I}_J R_W$ is the DC average voltage drop across the junction, and $\overline{I}_J = I_{DC} - \overline{I}_S = I_{DC} - \overline{V}_S/R_S$ the DC average junction current, respectively. The subgap resistance values extracted by this procedure are in reasonable agreement with extrapolations based on theory[37]. Finally, the critical current $I_c(V_g)$ is obtained using the voltage criterion $|\overline{V}_J| < 1.5\,\mu$V.

Figure 3a depicts a 2D map of the normalized microwave emission power at gate voltage $V_g = -0.08$ V, when the Fermi level is close to the bottom of the conduction band subband, and the critical current $I_c = 102$ nA. (We note that the position of the charge neutrality point shifts between cooldown cycles.) We observe prominent emission features at frequencies $f_J/2$, $f_J$, and $2f_J$, where $f_J = 2e\overline{V}_J/h$ is the Josephson frequency. In Fig. 3b, the RF amplitude at detection frequency $f_d = 5.16$ GHz, and the DC average junction current $\overline{I}_J$ are plotted against $\overline{V}_J$. The $\overline{I}_J(\overline{V}_J)$ trace has the expected shape for a Josephson junction that is loaded by an external shunting circuit. Additionally, it features three broad peaks that we attribute to an LC-resonance in the circuit (cf. refs. 19,21). By associating the peak positions with $hf_{LC}/ne$, $n \in \{1, 2, 4\}$, we extract the LC-resonance frequency $f_{LC} = 5.23$ GHz.

The presence of a large shunting reactance in the circuit affects the phase dynamics of a Josephson junction[18–22]. To explore the phase dynamics of our device, we model the experiment by the RCLSJ circuit C1, introduced in Fig. 2c. Here, the frequency-dependent microwave impedances of sample, wirebond connections, and surface-mount resistor are replaced by a small number of lump-circuit elements, and we disregard circuit loading by the biasing and detection branches as well as the gate. Finite temperature is modeled by adding white noise current sources, cf. refs. 19,38.

This value of $f_{LC}$ determines the product $L_S C_J$, but the ratio $L_S/C_J$ is yet unknown. To work this out, we analyze the shape of the $I–V$ curve

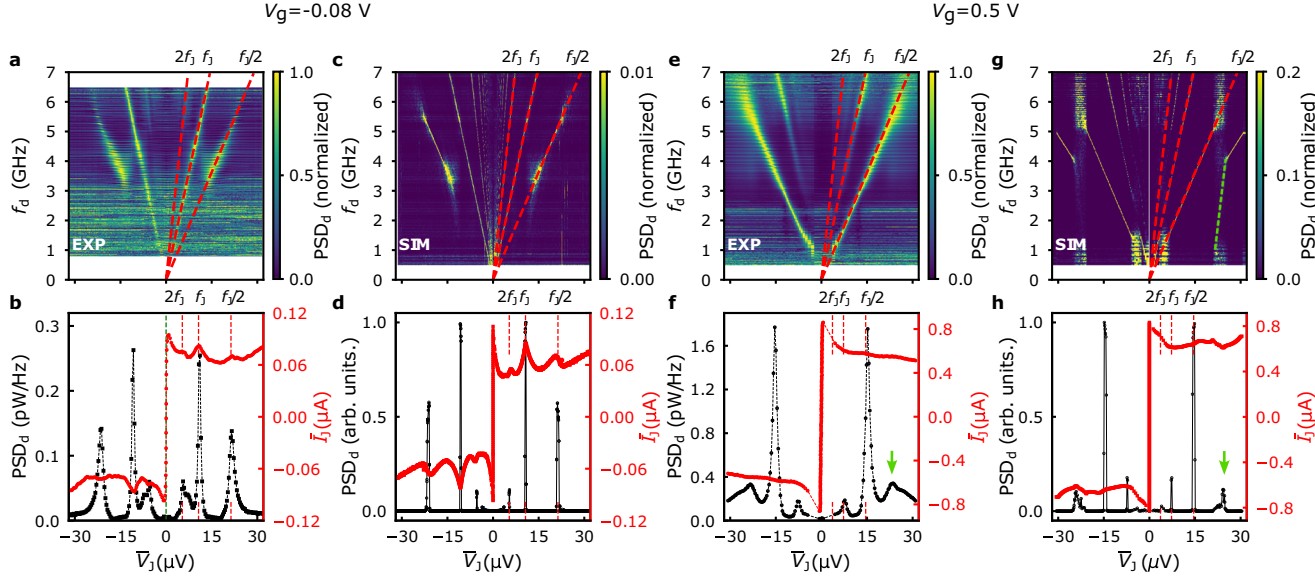

**Fig. 3 | Josephson emission and simulation. a** 2D map of the normalized power spectral density (PSD$_d$) as function of DC average junction voltage $\overline{V_J}$ and detection frequency $f_d$ at $V_g = -0.08$ V, close to the bottom of the conduction band subband. **b** DC average junction current $\overline{I_J}$ and PSD$_d$ as function of $\overline{V_J}$ at $f_d = 5.16$ GHz. **c** Fourier transform of $V_S(t)$ and **d** $\overline{I_J}(\overline{V_J})$ curve obtained by numerical simulation. The simulation parameters are $I_c = 102$ nA, $R_J = 444\,\Omega$, $C_J = 0.28$ pF, $L_S = 3.3$ nH, $R_S = 8.27\,\Omega$, $R_W = 10.23\,\Omega$, and $T = 7$ mK. **e** 2D emission map measured at $V_g = 0.5$ V when the Fermi level is high in the conduction band subband. **f** $\overline{I_J}$ and PSD$_d$ as

function of $\overline{V_J}$ at $f_d = 3.52$ GHz. **g** Fourier transform of $V_S(t)$ and **h** $\overline{I_J}(\overline{V_J})$ and PSD$_d$ obtained by numerical simulation. The simulation parameters are $I_c = 835$ nA, $R_J = 95\,\Omega$, and remaining parameters are identical to (**c**). The frequency range in which half-frequency Josephson emission occurs ($f_J/2$) and the LC-resonance features in the I-V curve are reproduced by the simulations. An additional emission feature (green arrow at $V_J \approx 21.5\,\mu$V in **f** and **h**) appears close to resonant bias ($hf_{LC}/e$) (green dotted line in **g**).

which is very sensitive to $L_S/C_J$ and run a series of numerical calculations of $\overline{I_J}(\overline{V_J})$ using a commercial circuit simulation software[39] and the SPICE circuit model of a Josephson junction[40]. Importantly, we disregard any microscopic aspects of the supercurrent transport affecting the current-phase relation of the device (cf. ref. 41) and assume:

$$I_X(\varphi) = I_c \sin\varphi \tag{1}$$

for all simulations, unless explicitly stated otherwise, where $I_X$ denotes the supercurrent, and $\varphi$ is the junction phase. The effect of other harmonics in the current-phase relation is discussed in Section S4 of the SI. The shape of $\overline{I_J}(\overline{V_J})$ in Fig. 3d matches the relative peak heights in the experimental data best (cf. Supplementary Information), yielding the parameter set $L_S = 3.3$ nH and $C_J = 0.28$ pF, which we subsequently use for all simulations in this circuit layout. The value of $L_S$ agrees well with a simple estimate based on the geometry of the bond wires ($\approx 3.4$ nH), and the I–V hysteresis in Fig. 1d is described well by the presence of the shunting capacitance $C_J$ (cf. Section S1 of the SI).

In Fig. 3c, g, we present the results of RCLSJ model simulations of the Josephson emission. For better visibility, we broaden the emission lines by introducing a small amount of white noise, corresponding to a noise temperature of 7 mK for the resistors in the circuit (Fig. 2c). A comprehensive analysis of noise broadening and a discussion of the linewidth of the emission features are provided in Section S4 of the SI. Figure 3c depicts a 2D map of the Fourier transform of the time evolution of $V_S$ (cf. Fig. 2a, c) as a function of $\overline{V_J}$ and $f_d$. The frequency range, where half-frequency emission ($f_J/2$) is predicted, closely matches the experiment. Indeed, the simulations reproduce the emission data and the shape of I–V curves well over a wide range of gate voltages; see Fig. 3e–h for a dataset at $V_g = 0.5$ V when the Fermi level is higher in the conduction band. The numerical simulation for the sample at larger density shows another emission line for $f_J/3$ at larger $\overline{V_J}$. For voltages close to resonant bias, $\overline{V_J} \geq hf_{LC}/e$, the feature diminishes and shifts to lower frequencies (green arrows in Fig. 3f, h and

green dotted line in Fig. 3g). The effect is observed more clearly in simulations with larger white noise (cf. Fig. S8 of the SI). The $f_J/3$ emission line is not clearly visible in the device we focus on in this paper. Yet, the shifted emission feature can be traced at lower frequencies close to resonant bias (green arrows in Fig. 3f, h). In addition, we present data on a further device in Section S5 of the SI, for which $f_J/3$ emission is clearly observed.

The agreement between simulations and experimental data suggest that the microwave emission at $f_J/2$ does not relate to an intrinsic property of the device but rather arises from period-doubling in the phase dynamics of the Josephson junction as a consequence of the sizable parasitic inductance in the circuit. The effect of shunt inductance and capacitance on the nonlinear dynamics of the junction phase has been topic of a wide range of theoretical work, analog simulation, and computer numerics [e.g., see refs. 18–24 and references therein]. A common approach to describing nonlinear dynamics problems is to determine manifolds of slow and fast dynamics in phase space[22]. The motion of the junction phase $\varphi$ follows (the stable branch of) the slow manifold. A shunt inductance $L_S$ folds this manifold, thus creating multiple stable and unstable branches with different $\varphi$. At extremal points, fast jumps between the stable branches occur. The distance $\Delta\varphi$ of the jump depends on the capacitance of the junction, conveniently specified by the dimensionless Stewart-McCumber parameter, $\beta = 2eI_cR_J^2C_J/\hbar$[35,36], where $\hbar$ denotes the reduced Planck's constant. For large enough $\beta$, phase evolution with $2n\pi$-periodicity and $n > 1$ becomes accessible. In the case of negligible resistance in the shunting branch of the circuit, $R_S \ll R_J$, the critical amount of folding to enable period-doubling can be calculated analytically. It is expressed by the inductance ratio $\alpha_c = L_S/L_J = 4.61$[22], where $L_J = \hbar/2eI_c$ is the Josephson inductance of the junction. Lowering the critical current is thus expected to eliminate phase trajectories with $4\pi$-periodicity.

We test this hypothesis experimentally by tuning the junction parameters in three different ways: applying gate voltage, magnetic field, or changing the temperature. The results are summarized in

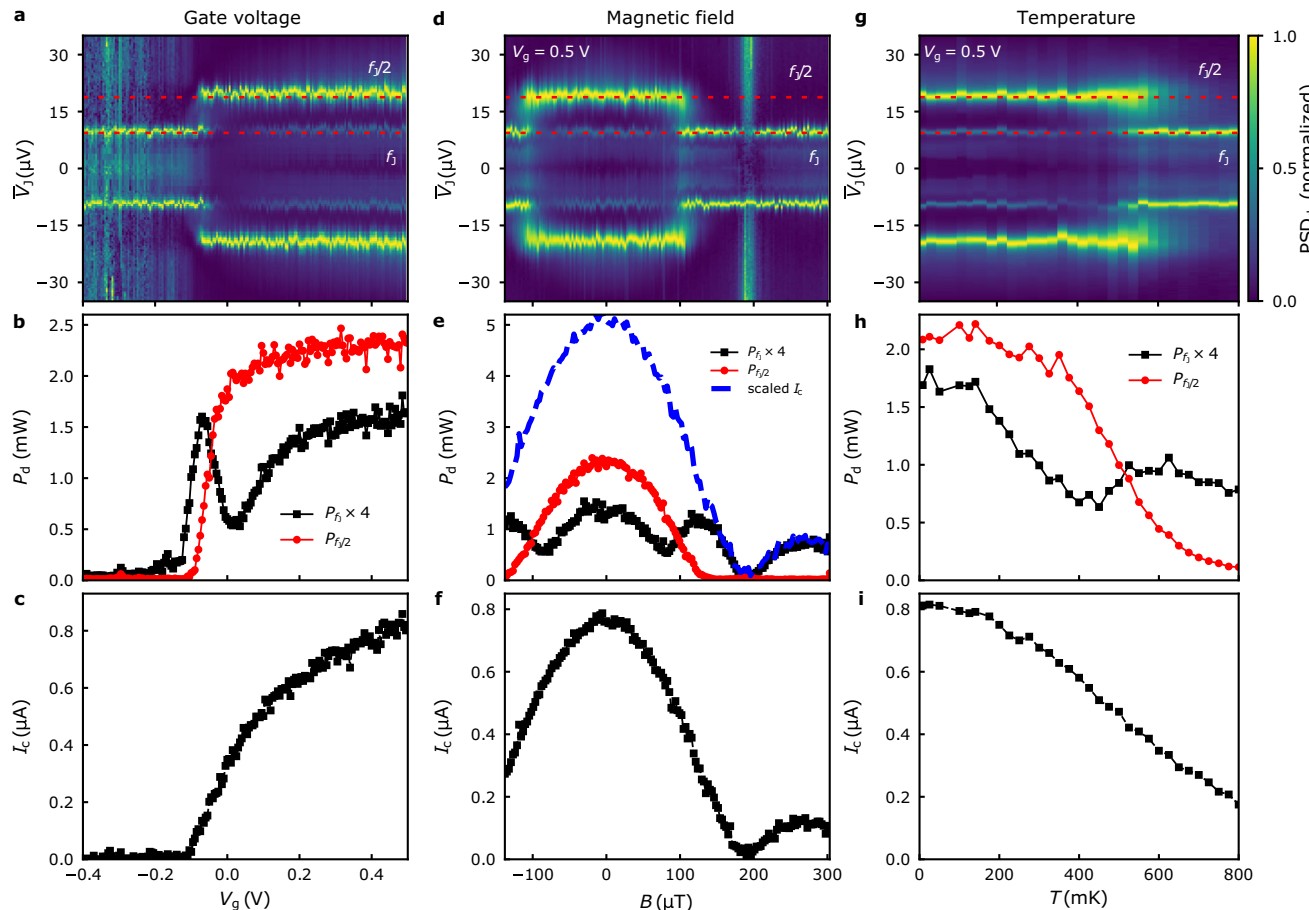

**Fig. 4 | Changing the dynamic regime.** 2D maps of the normalized power spectral density at detector frequency $f_d = 4.54$ GHz as a function of **a** gate voltage $V_g$, **d** perpendicular magnetic field $B$, and **g** temperature $T$, respectively, and the DC-averaged junction voltage $\overline{V}_J$. **b, e, h** The integrated power of the emission peak at detector frequency $f_d = 4.54$ GHz for $\overline{V}_J = 9.4\,\mu$V ($P_{f_J}$, black squares), and 18.8 $\mu$V ($P_{f_J/2}$, red circles), respectively. In the bulk gap, only $f_J$-emission is observed. **c, f, i** The critical current $I_c$ as a function of the external parameter. Measurements **d**–**f** are acquired at $T = 140$ mK to compensate for heating by resistive connections of the field coil.

Fig. 4. The data are plotted for fixed detection frequency $f_d = 4.54$ GHz at base temperature, unless indicated otherwise.

In Fig. 4a, the normalized power spectral density (normalized separately for each $V_g$) is mapped as a function of $V_g$ and $\overline{V}_J$. The color contrast allows to track the voltages $\overline{V}_J$, for which there is a strong microwave emission signal at frequency $f_d$. To quantify the emitted power associated with period-one and period-doubling dynamics, we define an approximate measure by integrating over the lineshape of the emission line $\text{PSD}_d(\overline{V}_J)$ centered around $\overline{V}_J = hf_J/2e$ [$hf_J/e$] and denote this quantity by $P_{f_J}[P_{f_J/2}]$, respectively; cf. ref. 13. At large negative gate voltages, we observe microwave emission at a single frequency $f_J$. As we increase $V_g$, the emission power $P_{f_J}$ at frequency $f_J$ exhibits a first upturn around $V_g \approx -0.13$ V but drops as soon as half-frequency emission ($f_J/2$) sets in Fig. 4b. The crossover happens around $V_g \approx -0.11$ V, concurring with a steep increase in $I_c$. By comparing the $V_g$-dependence of $I_c$ (Fig. 4c) with the DC characterization data (Fig. 1e), we find that the charge-neutrality point has shifted between the two measurements, and the crossover occurs close to the bottom of the conduction band subband. Importantly, we only detect emission at the Josephson frequency in the gate voltage region $V_g < -0.13$ V that we associate with the QSH insulator state. At higher $V_g$, the $P_{f_J}$ amplitude recovers. However, half-frequency emission ($f_J/2$) dominates, and $P_{f_J/2}/P_{f_J} \approx 6$.

We also study the magnetic field and temperature dependence of the microwave emission at $V_g = 0.5$ V (Fig. 4d–i). A perpendicular

magnetic field $B$ modulates the critical current. It follows a Fraunhofer-like diffraction pattern (Fig. 4f). The first node in the diffraction pattern occurs at $B = 180\,\mu$T. By contrast, the half-frequency emission, $P_{f_J/2}$, vanishes around $B = 130\,\mu$T and remains zero at higher fields (Fig. 4e). Whereas $P_{f_J}$ follows the shape of $I_c(B_z)$ at large $B$, we observe the same characteristic dip in the crossover region when $P_{f_J/2} \approx P_{f_J}(B = 0)$. The temperature dependence of the microwave emission follows a similar shape (Fig. 4g, h). As the temperature is increased, the emission power $P_{f_J/2}$ drops. There is a dip in $P_{f_J}$ at the crossover when $P_{f_J/2} \approx P_{f_J}(T \to 0)$. The dip feature is thus common to all three experiments in Fig. 4.

The results confirms the crossover to period-one dynamics when the supercurrent becomes small. The analysis is particularly simple for the external parameter $B$. In this case, only the critical current changes appreciably while other relevant parameters of the system remain approximately constant. We simulate the experiment by setting the critical current $I_c^{\text{sim}}(B) \equiv I_c(0)\sin(\pi AB/\Phi_0)/(\pi AB/\Phi_0)$, where $A$ denotes the effective junction area penetrated by the magnetic flux, and $\Phi_0 = h/2e$ is the flux quantum. The numerical result agrees with the experimental data qualitatively (Fig. 5). The crossover takes place in a region $|\alpha - \alpha_c| \lesssim 2$ (gray background in Fig. 5), centered around the critical value $\alpha_c = 4.61$[22]. A moderate increase in temperature rounds the shape of the $P_{f_J}$ and $P_{f_J/2}$ traces but does not change the width of the transition region substantially [cf. traces with open and solid symbols in Fig. 5].

As indicated above, we assume that the period-doubling phase dynamics is enabled by the parasitic inductance of the wirebonds. Therefore, we conduct a complementary experiment using the modified circuit C2. A small, low-inductance surface-mount resistor is glued on-chip with a conducting silver-epoxy glue to connect between the bonding pads. The Josephson junction is shunted by a total shunt resistance $R_S = 27.2\,\Omega$ and connected to the external circuit via a series resistance of $R_{ser} = 10.5\,\Omega$. We find that the gating efficiency has

changed, however, the extracted subgap resistances and critical current values mutually match with the data of circuit C1 and the unshunted measurement with reasonable accuracy.

The data for circuit C2 are summarized in Fig. 6. There is no indication of period-doubling dynamics: we observe a strong microwave emission signal at the Josephson frequency ($f_J$) (Fig. 6a–c, e). Additionally, a faint contrast is present at $2f_J$. There are no distinct LC-resonance features present in the $\overline{I_J}(\overline{V_J})$ traces (Fig. 6b). And lastly, the emission power $P_d$ scales with the critical current $I_c$ as gate voltage or magnetic field are varied (Fig. 6c–f). The data are plotted for fixed detection frequency $f_d = 5.2$ GHz at base temperature, unless indicated otherwise.

Our consistent modeling of period-doubling in the DC-biased RCL-shunted junction circuit obviously raises the question how the emission results relate to missing Shapiro steps in the $I–V$ characteristics of the AC-driven junction. We indeed observe a suppression of odd Shapiro steps in the unshunted device (Fig. 1f); and the steps re-emerge when a magnetic field of similar magnitude is applied as in Fig. 4d, e. This measurement has been carried out using a home-built sample holder with a lead-less chip carrier (LCC) system. The sample is wirebonded to the chip carrier. A coarse estimate of the microwave impedance between the sample holder leads yields $|Z| \lesssim 100$ V A$^{-1}$, suggesting that the electromagnetic environment of the junction is dominated by the parasitic inductance of the wirebonds [cf. Section S1.2 of the SI for a detailed discussion]. We thus repeat the Shapiro step experiment in circuit C2 (Fig. 7). In this circuit layout, the device is resistively-shunted without adding the large parasitic inductance of wirebonds. We map out the Shapiro steps for several frequencies by coupling microwaves into the circuit via the top-gate line. Only conventional Shapiro patterns with voltage steps at $V = n \times hf/2e$, $n = \pm1, \pm2, \ldots$, are observed (Fig. 7a–d). Although preliminary, our results on Shapiro steps hint at excess parasitic inductance as a possible cause of period-doubled phase dynamics in such experiments.

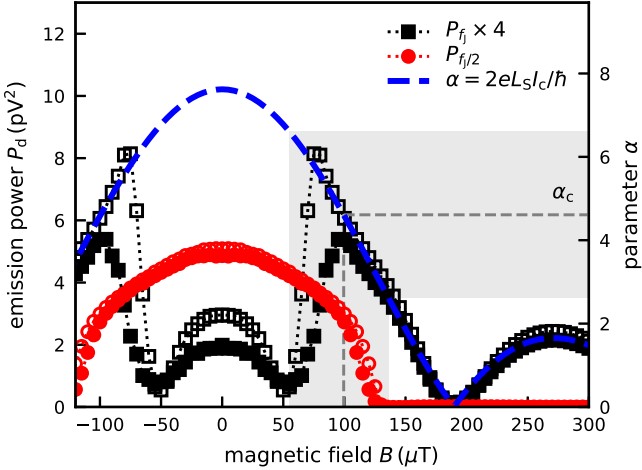

**Fig. 5 | Simulation of the emission power with an applied magnetic field.** The frequency-integrated power of the emission peak $P_d$ at frequency $f = 4.54$ GHz for $\overline{V_J} = 9.4\,\mu$V ($P_{f_J}$, black squares) and 18.8 $\mu$V ($P_{f_J/2}$, red circles) as function of perpendicular magnetic field $B$. At large $B$, the half-frequency emission is zero, and $P_{f_J}$ is proportional to $I_c^{sim}(B)$. The dashed blue line shows the value of the parameter $\alpha = 2eL_S I_c/\hbar$. The simulation parameters are $I_c^{sim}(0) = 760$ nA, $T = 140$ mK (solid) and $T = 10$ mK (open symbols). The remaining circuit parameters are identical to Fig. 3g, h.

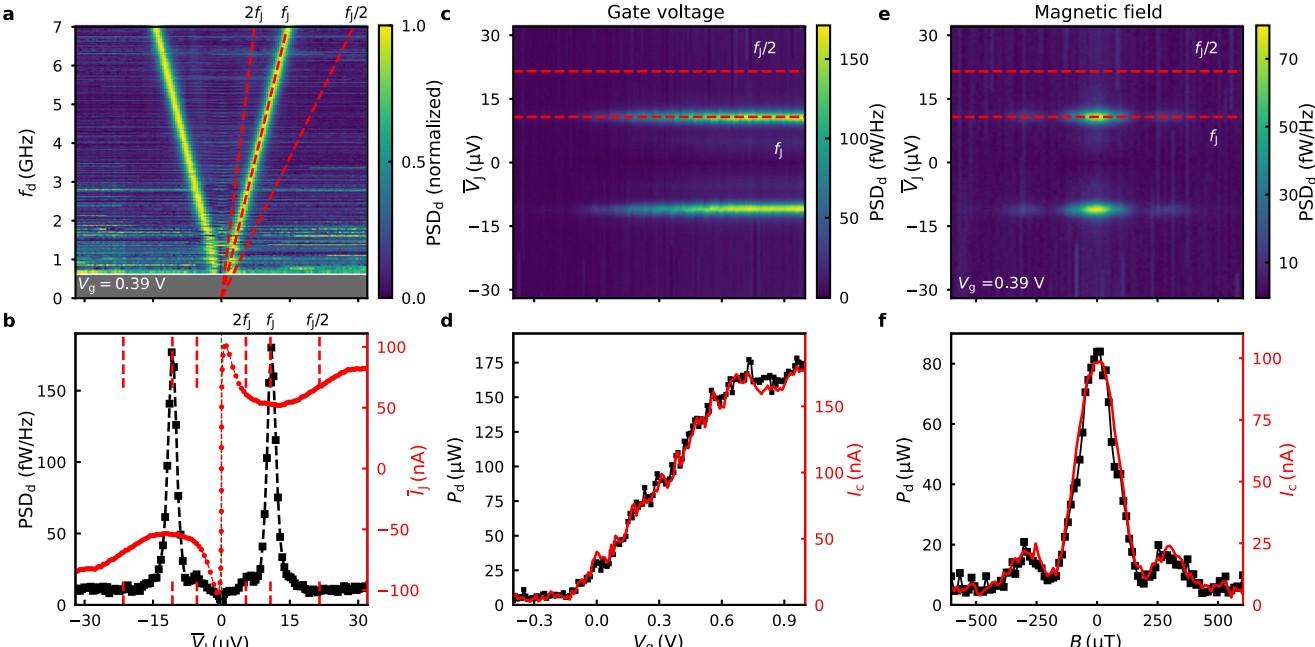

**Fig. 6 | Emission experiment with on-chip shunt. a** Normalized power spectral density as function of DC-averaged voltage $\overline{V_J}$ and detection frequency $f_d$ at gate voltage $V_g = 0.39$ V. **b** DC-averaged junction current $\overline{I_J}$ and power spectral density as function of $\overline{V_J}$ at fixed $f_d = 5.2$ GHz. 2D maps of the power spectral density as function of $\overline{V_g}$ (**c**) and perpendicular magnetic field $B$ (**e**) for $f_d = 5.2$ GHz. The

integrated power of the emission peak at frequency ($P_{f_J}$, black dots) and the critical current $I_c$ (red line) are plotted as function of $V_g$ (**d**) and $B$ (**f**) for $f_d = 5.2$ GHz at $V_g = 0.39$ V. Measurements **e** and **f** are acquired at $T = 150$ mK to compensate for heating by resistive connections of the field coil.

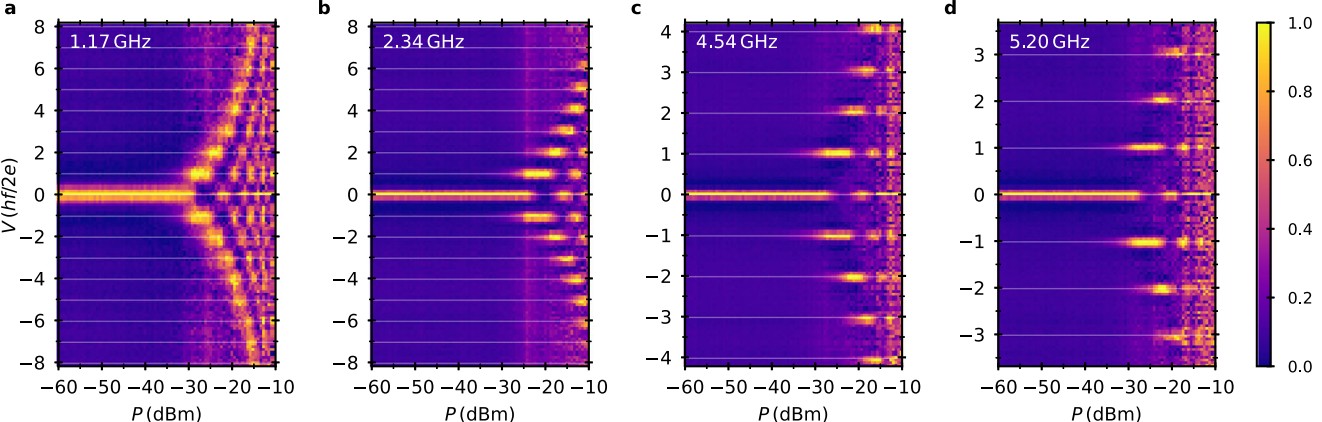

**Fig. 7 | Shapiro step measurements on the shunted device (circuit C2).** Voltage histogram of the junction current as function of microwave power $P$ and voltage drop $V$ across the junction at gate voltage $V_g = 0.39$ V and microwave frequency **a** $f = 1.17$ GHz, **b** $f = 2.34$ GHz, **c** $f = 4.54$ GHz, and **d** $f = 5.2$ GHz, respectively. After subtracting the minimal bin count, we normalize the bin count difference by dividing by the maximum bin count difference for each value of $P$ to achieve best visibility as the current counts distribute unevenly due to the non-monotonicity in the I-V characteristic. In circuit C2, Shapiro steps at all multiples of $hf/2e$ are observed.

## Discussion

Josephson junctions with novel weak link materials may operate in parameter ranges that are not typically encountered in conventional Josephson junction architectures. The design consideration of such devices include requirements that are foreign to standard SIS- and SNS-junction technology (e.g., low-thermal-load lithography processing, poor material adhesion, substrate and film strain, etc.) and require trade-offs in the sample geometry and metal lead wiring that affect the dynamics of the system, as we demonstrate above.

In the present device, the large shunting capacitance mostly stems from the metal lead wiring layer that is deposited directly on the CdTe substrate, which has a dielectric constant $\epsilon_r \approx 10$. This changes the junction dynamics and causes a hysteresis in the $I-V$ characteristics (see Fig. 1d and refs. 3,4) Additionally, the wirebond connections to the external measurement circuit provide considerable parasitic inductance. As, for typical HgTe QW Josephson junctions, the critical current and normal state resistance vary by two orders of magnitude when moving the Fermi level from high in the conduction band subband into the QSH regime, the inductance ratio $\alpha$ and the capacitance parameter $\beta$ change over a wide range. Thus we encounter different regimes of nonlinear dynamics in a single device. Previously, theoretical analyses focused on the effect of the shunting capacitance on the observability of an intrinsic $4\pi$-periodic supercurrent; e.g., see refs. 42,43. Alternative mechanisms of period-doubling dynamics have not been systematically explored in this context. Our experiments demonstrate that experimental signatures such as $f_J/2$ emission and suppressed Shapiro steps should not be taken as a reliable indicator for the presence of the fractional AC Josephson effect, unless the junction phase dynamics are carefully analyzed and the RF environment is known.

We stress that the very simple modeling, we performed above, is sufficient to explain all salient features of the emission experiment. We make no assumptions about microscopic properties of the device (e.g., intrinsic $4\pi$-periodic supercurrent or other harmonics in the current-phase relation, bias dependence of the subgap conductance, or driven transitions between Andreev bound states). Also, other aspects of the microwave circuit, such as the transfer function of the detection circuit, microwave losses via the gate connection or in the substrate, and circuit loading by the biasing and detector connections, have been omitted from the discussion for simplicity. These aspects of the experiment are important only for determining absolute magnitudes of the detected microwave signals. We have carried out additional numerical simulations exploring the effect of white noise fluctuations,

current-phase relation and circuit-loading by the detector [cf. Section S4 of the SI]. The numerical results ascertain that our interpretation of the experiment remains robust for a wide parameter range.

Our work highlights the need to analyze the phase dynamics before designing an experiment that studies the fractional AC Josephson effect. This involves aspects beyond the lithographic design of the sample; e.g., resonant modes of the RF measurement circuit and radiative coupling to cavity modes in the sample fixture have to be considered. Additionally, the frequency ranges for the experiment and the quality factor of the phase oscillations have to be chosen carefully if one intends to detect a small $4\pi$-periodic supercurrent component in a sample with moderate to large $2\pi$-periodic supercurrent[43,44]. To our best knowledge, the effect of other components in the current-phase relation on the detection of the fractional Josephson effect have not received as much attention in the literature but may be crucial for interpreting Shapiro step patterns[45] or the presence versus absence of Josephson emission lines at a given frequency. A complementary measurement of the current-phase relation (or a fast switching current distribution measurement[46,47]) may be necessary to obtain clarity.

Several ways exist to improve the experiment: a well-defined RF environment is achieved by connecting the sample on one end of a co-planar waveguide and locally shorting the Josephson junction with a microstructured thin-film resistor that is matched to the impedance of the transmission line. This imposes some practical limitations regarding the value of shunt resistance[48]. Alternatively, a capacitively-coupled Josephson junction or DC SQUID can be used as an on-chip detector[14,15], if possible to realize lithographically, which eliminates the need to establish a high-bandwidth connection to an external RF circuit. The detection principle is based on photon-assisted tunneling. Still, the phase dynamics of the test device needs to be damped sufficiently to probe for $4\pi$-components in the supercurrent. In refs. 15,17, this is achieved by coupling the junction to a broad environmental resonance instead of a discrete shunting resistor. As the frequency-dependence of the environment enters into the expressions for the detector current and noise fluctuations, again, careful engineering of the device parameters is required. As the DC SQUID represents an inductive load, it should be included in the analysis of the phase dynamics of the junction.

We conclude by remarking that the observation of period-one phase dynamics (i.e., a $2\pi$-periodic Josephson effect, or $f_J$-emission) is

the theoretically predicted, experimental outcome for QSH Josephson junctions in the absence of time-reversal-symmetry (TRS)-breaking[5], dissipation-enabled $4\pi$-periodicity[6], or an equivalent mechanism. Aside from breaking TRS by introducing a local magnetic interaction, restoring the $4\pi$-periodicity of the AC Josephson effect of topological Josephson junctions requires careful tuning of device parameters, conditions unlikely to be fulfilled accidentally. In light of our findings, a careful reevaluation of claims regarding the observation of a fractional AC Josephson effect is warranted. Unfortunately, the present layout of our experiment does not allow for the implementation of TRS-breaking by application of a local magnetic field. Future work will focus on this aspect.

In summary, we have conducted microwave emission measurements on a topological HgTe quantum well Josephson junction in two different wiring configurations. The circuit with larger wiring inductance exhibits period-doubling in the phase evolution of the junction, and the voltage across the shunted device oscillates at half of the Josephson frequency ($f_J/2$). We detected the phenomenon by directly recording the microwave emission of the circuit. Next, we numerically simulated the nonlinear dynamics of the system using a simple RCLSJ circuit with a $\sin\varphi$ current-phase relationship. This model captures all essential experimental observations. Therefore, we conclude that the emission signal at $f_J/2$ originates from a period-doubling induced by shunt inductance. It does not constitute evidence for an intrinsic $4\pi$-periodic component in the current-phase relationship of the Josephson junction. The absence of microwave emission at $f_J/2$ in a complementary experiment, carried out in a low-inductance configuration on the same device, supports this interpretation. Our observations demonstrate that semiconductor Josephson devices operate in a parameter regime for which careful modeling of the electromagnetic environment becomes essential for interpreting the response. This must be taken into consideration when searching for topological properties in the dynamics of Josephson junctions with topological insulator weak links.

## Methods

### Sample fabrication
The experiments are conducted on a HgTe QW Josephson junction in a side-contacted device geometry. The device is fabricated from a band-inverted $Cd_{0.7}Hg_{0.3}Te/HgTe/Cd_{0.7}Hg_{0.3}Te$ quantum well heterostructure (Q3278, device ID: Q3278-2) with a 8.5 nm thick HgTe layer, grown by molecular-beam epitaxy. The as-grown material has a carrier density of $0.79 \times 10^{11}/cm^2$ with a carrier mobility of $1.23 \times 10^5 cm^2/(V\,s)$, determined in a Hall effect measurement on a large Hall bar device, fabricated from the same wafer. Using a self-aligned wet-etching and deposition process, a $4\,\mu m \times 0.2\,\mu m$ mesa structure is shaped, and superconducting Ti/Nb/Ti/Au electrodes are sputtered at an angle to cover the side-walls of the mesa (Fig. 1a). The film thicknesses of Ti and Nb are ~ 5 nm and ~ 100 nm, respectively. Additionally, top layers of 3 nm Ti and 5 nm Au prevent oxidation of the niobium. A scanning electron micrograph of a side-contacted mesa is shown in Fig. 1b. In the last step, a top-gate is fabricated by depositing (using atomic-layer deposition) a ~ 14 nm hafnium oxide dielectric layer and a Ti/Au metal stack ( ~ 100 nm thick) as gate electrode.

### RF sample mounting
For the measurements in Figs. 3 and 4, we use a commercial surface-mount resistor for shunting. The solder coating is stripped from the terminals to improve adhesion of the wirebonds. Sample and bias resistor are glued to a copper-backed printed-circuit board (PCB) header. One terminal of the shunt resistor $R_S$ is wirebonded to DC bias line and bias tee connector, the other terminal to ground. The Josephson junction is wired in parallel by wirebonding directly to the terminals of the shunt resistor (Fig. 2a). The RF sample header is placed in a RF-shielded box equipped with a small superconducting solenoid.

The box is mounted inside a magnetic shield in the sample receptacle of a dilution refrigerator with fast-loading mechanism and base temperature 7 mK.

### Measurement circuit
A schematic of the measurement setup is given in Fig. 2a. The bias current ($I_{DC}$) is supplied using the voltage output of a D/A converter ($V_{DC}$) in series with a 1 M$\Omega$ resistor and an RC-filter with a corner frequency of 1 Hz. Coax lines for DC signals are filtered by commercial LC low-pass filters at base temperature. An amplifier cascade provides a total RF signal gain of ~ 100 dB. The first stage is a cryogenic low-noise amplifier (LNF-LNC4_8C, LTA) attached to the 2 K plate of the dilution refrigerator. Further amplification is provided by two amplifiers at room temperature (RTA). The spectra are recorded by an HP 8563E spectrum analyzer. Density maps are generated by setting the spectrum analyzer to detecting the power $P$ in a narrow frequency band with center frequency $f_d$ and bandwidth $\Delta f = 2$ MHz and sweeping the bias voltage. Power spectral density values are calculated by dividing the detected power by the detection bandwidth, $PSD_d \equiv P/\Delta f$.

## Data availability
The data supporting the findings of this study are available at https://doi.org/10.5281/zenodo.14894222.

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

## Acknowledgements

We thank Erwann Bocquillon, Russell Deacon, and Anton Akhmerov for valuable discussions. We gratefully acknowledge the financial support of the ERC-Advanced Grant Program (project "4-TOPS", grant agreement No. 741734 [L.W.M.]), the Free State of Bavaria (the "Institute for Topological Insulators" and the "Munich Quantum Valley Initiative", supported by Hightech Agenda Bayern Plus, StMWi-43-6665r-2/11/3), and the Deutsche Forschungsgemeinschaft (DFG, German Research Foundation) - SFB 1170 (Project-ID 258499086) and EXC2147 "ct.qmat" (Project-ID 390858490) [C.G., J.K., H.B., M.P.S., L.W.M.].

## Author contributions

W.L., S.U.P., X.L., S.U., L.F., C.G., J.K., H.B., M.P.S., and L.W.M. planned and designed the experiment. L.F. grew the material, and X.L. fabricated the HgTe devices. W.L., S.U.P., and S.U. performed the experiments. W.L. and M.P.S. carried out the numerical simulations. All authors participated in the analysis led by W.L., S.U.P., and M.P.S. All authors participated in writing of the manuscript.

## Funding

## Competing interests

The authors declare no competing interests.
