## [Transparent Peer Review file · Nature Communications]

Period-doubling in the phase dynamics of a shunted HgTe quantum well Josephson junction

Corresponding Author: Dr Martin Stehno

Version 1:

Reviewer comments:

Reviewer #1

(Remarks to the Author)

This paper by Wei Liu et al. presents their work related to the microwave emission by a Josephson junction based on a HgTe heterostructure placed in an electromagnetic environment. The paper focuses on the different radiation lines that can be observed in this system in relation with how the different elements placed in parallel to the main junction influence the microwave emission by making new lines appear at half the Josephson frequency of the system. Because the system does not meet the necessary topological conditions, such doubling of the periodicity in the emission cannot have a topological origin. Therefore, the authors make the point that previous results obtained in such system should be re-examined in the framework of an emission modified by the electromagnetic environment of the junction.

I would like to start my review by saying that I believe that this type of paper is of a broad interest and valuable to the community as it brings alternative interpretation to sometimes sensationalistic papers. I find that this paper contains interesting results that might be worth being published in Nature Communications. However, I have several remarks and questions for the authors. In particular I find the paper hard to read, especially in relation to the notations used and the quantities measured that would gain from being explained more clearly. I would advise the author to extend the supplementary materials section so that the readers of the manuscript could have access to more complete data sets and analysis methods. In particular the data that have been renormalized are missing a dedicated section to elaborate on the normalization methods that were used. In the current state I cannot recommend this paper for publication in Nature Communications.

Below I list my remarks that I hope the authors can answer.

1. In the introduction, the authors discuss TRS-breaking by a (local) magnetic field. At this point I have a question about the local field. Is it worth mentioning it at this point in the text? From what I understand, a small superconducting coil was used to generate the magnetic field. I would suggest, if this mention has no influence on the rest of the discussion, to get rid of this parenthesis.
2. In page 4, the sentence "The numerical analysis suggest that the system is close to a period-doubling instability, causing the 4π periodic evolution of the junction phase ϕ ." Mentions a numerical analysis. This analysis is nowhere to be found in the manuscript. I would expect that it at least appears as supplementary material and be discussed more thoroughly.
3. In figure 1.b, a scale bar is missing to indicate the size of the structure. At minima it should appear in the caption of the figure.
4. In figure 1.f&g arrows are presented to indicate the $n=1$ & $n=3$ missing steps. However, they are not mentioned in the caption. I believe it would make the figure clearer to add a mention in the caption.
5. In fig.2, the schematic of the measurement circuit indicates a voltage V_{DC} (fig1.a) and another voltage V_S that seem to both connect to the junction/the shunting circuit. From what I understand, the voltage that is applied is V_{DC} that generates a current I_{DC} and V_S is the measured voltage down at low temperature that is used to extract the critical current. Similarly, V_J , if I understand correctly is the measured voltage across the Josephson junction that will be dependent on the shunting resistor and the rest of the circuit. Finally, in the text the notation \bar{V}_J is also used. What is the meaning of the averaging here? Generally, I think the author should do a better job at clearly defining the quantities they use in their work in order to avoid any confusion on the reader's side.
6. In page 6, the page begins with "At $B=0$, the current-biased device exhibits the hysteretic current-voltage (I-V) characteristic of an undershunted Josephson junction. I have two comments in this instance. The first one may be personal due to a gap in my knowledge but I don't understand the term "undershunted". I understand what the shunted par comes from but "undershunted" seems to come in opposition to "overshunted" or "normally-shunted". What is the criterion for being undershunted? Also looking at refs. 33 & 34 I couldn't find this expression used either. Could the authors clarify what they

mean by “undershunted”? My second point is related to the $B=0$ statement. Could the author present (probably best in the supplementary) curves at $B \neq 0$? How does the hysteresis evolve at finite field?

7. Later in the same paragraph, the main claim of the paper appears as the first and third steps of the Shapiro steps are missing in fig.1.f, thus mimicking what is expected for a fractional system. This is indeed true at low power, but I can see traces of these steps at higher power. Could the authors explain the role of the power and compare to what is expected for a fractional system? Also, from this data I imagine that the authors have at their disposal dI/dV maps as a function of voltage and power. I think it would be helpful to include some of those maps in the supplementary to have a complementary way to look at the data and maybe even make some comments on the comparison with the expected behavior in topological systems.

8. Page 6 again, it is written that the fridge has a base temperature of 7 mK. However, earlier in the results section it is mentioned that the base temperature is 35 mK. Please clarify this point.

9. In fig. 3.g I see observe another resonance that appears in the simulation that is not observed in the experimental data. Does it correspond to $f_j/3$? $f_j/4$? Could the authors comment on this?

10. Concerning the simulations in figure 3, it is said in the text that finite temperature is simulated by adding white noise current sources in parallel to the resistors (p9). My naïve guess would be that this has the effect of broadening the resonances such as those seen in fig3. My question is related to the amplitude of this white noise. My feeling is that it is underestimated as the width of the resonance in simulations in fig. 3 is very small compared to the experimental data. Are there other mechanisms that could explain this additional broadening. What does the simulation look like for various values of the noise amplitude?

11. In fig. 4 I find it hard to understand what is plotted in figures b, e and h. From what I understand figure a, d and g correspond to the emission as a function of \bar{V}_J and the gate voltage (hence the density) and the z axis correspond to the power detected at the frequency f_d (which is not specified). I would tend to think that $f_d = 4.54$ GHz as it is the same frequency used in b, e and h. However, I am failing to understand the link between the first and second row figures. If b, e and h are simply cuts from a, d and g along the dotted red line I don't understand the data that is plotted as the color code coded data in the map doesn't have the same evolution as a function of the various parameters as the b, e and h lines. Does it have to do with normalization procedure? Then it should be explained more clearly. Could the author help me understand better the quantities plotted in these figures? What do the author mean by “the integrated power of the emission peak” precisely. I find it confusing and that this harms the readability of the paper. In short, could the author make it clearer what is the relationship between P_{f_j} and the spectral density PSD_d ?

12. In figure 4.e the authors plot the field dependence of the critical current on top of the integrated power and compare these two quantities, in particular insisting on the similarity of the shape of the P_{f_j} curve at large field that follows that of the critical current. Isn't there a claim that could be made here that the sum of the emission at P_{f_j} and $P_{f_j/2}$ reproduce somehow the whole field dependence of the critical current or at least they are link in some way. I also find the explanation of the link between $P_{f_j/2}$ and I_c to be very qualitative but missing a bit of physics to get a sense of why these two quantities are linked and the underlying phenomena.

13. In page 10 the second to last paragraph starts with “Fig. 3c depicts a 2D map of the Fourier transform...”. This data is a simulation; however, I find that it is not apparent from the beginning. I would advise the author to make it clear when they are presenting a simulation vs. actual measured data.

14. Page 11 the second line mentions “a wide range of theoretical work, analog simulation, and computer numerics” but no citation accompanying this sentence. It would be worth citing some of these works or reviews.

15. Page 12, in the sentence “it follows a Fraunhofer-like diffraction pattern” referring to fig.4f, I agree with the statement. However, this behavior is better seen in fig. 6 e & f. I would advice this figure to be mentioned here as well or maybe that the Fraunhofer discussion comes a bit later so that fig 6 had already been introduced.

16. Page 13, the first full paragraph concludes with a comment on the effect of the temperature increase on the features of the P_j . I guess this means that the authors have this data and I think they should consider adding it as a section to the supplementary materials.

17. In figure 6, it seems to me that there is a small spectral weight that is barely visible in the PSD maps. Maybe the authors should discuss that at the end of page 13 as they explicitly write “we observe microwave emission only at the Josephson frequency (f_J)”.

18. In figure 7, the caption of the figure reads “the data are normalized for best visibility”. At this point my feeling is that more details about the normalization procedure are needed. Is the normalization global or at constant power or constant voltage?

19. Page 16, the first paragraph concludes with “We have carried out additional numerical simulations for realistic circuit-loading conditions, finding no qualitative changes in the dynamics”. Where are those numerical simulations? Shouldn't they appear in the supplementary to support your claim?

Response to request for comment on report of Reviewer #3

1 - From what I understand from the Nature communication by Laroche et al., the authors study the 4π Josephson effect in InAs nanowires with a wire coupled to a broad band detector. For me the paper under review and that paper address different aspects of the problem. The Nature communication follows the path trying to identify clear signatures of the 4π Josephson effect while taking into account the electromagnetic environment of their Junction. However, this paper lacks full time simulations that would allow to compute the expected shape of the $I(V)$ characteristics and study the potential emergence of non-topological 4π periodicity. In contrast the paper under review examines the case of a system NOT in a topological configuration that nevertheless shows signs of 4π periodicity. Therefore the paper is trying to prove that the simple disappearance of Shapiro steps cannot be considered as proof of a topological behavior of their system. I agree with the referee that a comment of the author about this work and at the very least a citation would be necessary in order to draw a complete picture of the recent experiments on the subject. Therefore I think that this paper brings enough new information

compared to the Nat. Comm. paper to guarantee publication (as the referee says, whether or not this should be in Nature communications remains to be seen after the authors answer the first round of comments from the referees).

2 - Relating to the reference [19] of the paper under review. I agree with the referee that the principle of the two papers are quite similar. However, the 1998 paper does not study the 4π periodic Shapiro steps per say. It does not exclude its appearance explicitly but fails to comment on that. For this reason I think the Liu paper pushes further the ideas of [19] applying them to a very specific situation. Therefore I don't think it would be so straightforward to get to the result of the submitted paper solely based on [19].

3 - Once again in my understanding of the Laroche paper, the scope of the work is quite different despite the thematic overlap and the take-home message of both papers are quite different. As I said in my first answer I think a decision regarding the adequacy of Nat. Comm. for publication remains to be decided after the authors answer the questions of the referees.

Reviewer #2

(Remarks to the Author)

In "Period-doubling in the phase dynamics of a shunted HgTe quantum well Josephson junction" Liu et al. study Shapiro steps and microwave emission from a HgTe Josephson junction. They observe a missing Shapiro step at low magnetic field, similar to [3,4]. They observe subharmonic ($fJ/2$) Josephson radiation at low magnetic field, low temperature, and not-too-negative gate voltage. Their findings are explained by the nonlinear dynamics of a well-motivated circuit model. Moving to an on-chip shunt resistor removes the $fJ/2$ radiation and restores all Shapiro steps, consistent with the model.

There have been many works associating missing Shapiro steps with topological phases. It is generally understood that there are several non-topological reasons that these can occur, as the authors discuss in their introduction.

This work is to my knowledge the first to deeply understand the origin of the missing step, and to be able to restore the expected "textbook" behavior. It therefore plays the role of clearing up questions that linger in the old literature, and presenting a potential path forward for using Shapiro steps and Josephson radiation as a meaningful diagnostic of topological superconductivity.

This contribution is significant enough to appear in Nature Communications.

I do not have major scientific concerns about the work. A few minor questions that I would like the authors to address:

(1) The authors identify hysteresis in their current-voltage characteristics as being from underdamped phase dynamics. I am skeptical that this identification can be made. Have the authors critically assessed that this could be from overheating? A simple estimate would be to estimate the effective equilibrium temperature T^* in the normal state accounting for Joule heating, and compare the observed "re-trapping" current with $I_c(T^*)$.

(2) Can the authors supply circuit parameters for their on-chip resistor solution? I am asking for circuit parameters including stray inductance/capacitance, not merely the nominal lumped-element resistance. Could this be a practical way of realizing a high impedance environment for dual Shapiro physics and the like?

Response to request for comment on report of Reviewer #3

I do not think the mere existence of Laroche et al. is a major challenge to the novelty or significance of the present work. However, I am quite surprised that it was not cited and discussed in a clear way. Referee 3 has done a service by pointing this out. It would be interesting to hear why the authors did not cite this work.

I think Ref. 19 is cited appropriately. A reasonably skilled expert would certainly take away from Ref. 19 that dynamics are complex and need to be thought about carefully. Certainly this has been appreciated. But knowing that some previous $fJ/2$ observations were due to period-doubling dynamics, which can be completely suppressed with an on-chip resistor, is not possible just from reading [19].

Properly citing and discussing Laroche et al. will most likely alter this work, but in my opinion it is unlikely to compromise novelty and impact.

Citing and discussing Laroche et al. will cause the authors to deal directly with the fact that prior works have indeed considered junction dynamics more carefully. On the other hand, I am not sure to what extent Laroche et al. really engineered their environment, as opposed to fitting for effective environmental RC values that give overdamping (and are consistent with earlier work pioneering their technique). I do find it questionable if the RC parameters Laroche et al. determine are really correct. It seems plausible that the capacitance of their Al/AIOx test junction is part of their RC fit, but that this is essentially absent in their real InAs device. This would lead to a very different environmental impedance than what is modeled, possibly altering the phase dynamics. This makes it all the more valuable to see that the environment can be directly controlled with an on-chip resistor, and that this is definitely needed in some approaches.

Reviewer #3

(Remarks to the Author)

Dear Editor,

The manuscript titled "Period-doubling in the phase dynamics of a shunted HgTe quantum well Josephson junction" compares extensive RF measurements of emitted radiation from a HgTe Josephson junction (JJ) with careful microwave simulations to show that the $fJ/2$ frequency radiation peak often associated with topological Josephson junctions, notably in 3 previous papers (Refs. 3,4 and 14) by the present group can be attributed to non-linear dynamics from circuit resonances similar to those studied in trivial JJs in Ref 19. Apart from several theoretical works cited in this manuscript - related claims have been made for the doubled Shapiro step (Refs 11 and 12), which was motivated by Ref 1 in the nanowire. The present manuscript seems to be the first to compare fractional Josephson radiation from a possibly topological JJ to simulations of such radiation from a trivial JJ (a la Ref 19) and show that this is trivial. Thus, this work apart from shedding light on previous experiments from this group i.e. Refs 3,4 and 14 can potentially provide guidance for the field of topological JJs - though the phenomenon reported here is essentially the physics reported in 1998 (i.e. Ref 19).

While I agree with the findings of this manuscript and consider them significant enough to be published whether they should be published in a journal for a broad readership depends on the level of guidance this manuscript can provide to the field of topological JJs beyond Refs. 3, 4 and 14. Radiation measurements, though preferable are difficult and therefore there are not a lot of experiments in this area. In this regard it is notable that Nature Communications volume 10, Article number: 245 (2019), which claimed to observe fractional JJ radiation in nanowires, is not cited in the present manuscript. In fact, this paper claims

"This on-chip detector²⁹, coupled via capacitors CC to the NW junction (see Fig. 1b for the schematics and Fig. 1e for an optical image of the device), was engineered to result in an overdamped microwave environment characterized by a single $f_c = (2\pi RC)^{-1} \approx 28$ GHz cutoff frequency with $R = 538 \Omega$ and $C = 10.4$ fF (see Supplementary Fig. 2). The resulting broadband coupling to the detector⁷ inhibited higher order photon emission, which could mimic the 4π -periodic Josephson effect^[30]." where Ref 30 is Cawthorne et al (Ref 19 in this work).

Specifically, they claim to have ruled out the effect discussed in this work. I think it is crucial for the present manuscript to discuss this claim to some degree, which is of course limited given that the authors may not have access to the relevant data from this paper to determine the validity of the claim.

The present manuscript suggests: "We stress that the very simple modeling, we performed above, is sufficient to explain all salient features of the emission experiment". However, it is not clear that this would be applicable to many topologically trivial JJs. For example it is unclear if such modeling is feasible when the current-phase relation is more complicated which is for example the case even with many topological materials Phys. Rev. Lett. 114, 066801 – Published 9 February 2015 and Phys. Rev. B 100, 064523 (2019). Do the authors suggest determining the current-phase relation in some independent way? This would be good for the manuscript to discuss.

A minor point is that the discussion of circuit details at the bottom of page 7 seems to present details that are difficult to follow and not obviously relevant for a general reader. Maybe the authors should consider moving this discussion to a different section i.e. methods or Supplement.

Another relatively minor point, in the context of the discussion of radiation "detection of microwave photons emitted by the voltage-biased Josephson junction" is that Ref 10 explicitly calculates the $fJ/2$ radiation peaks from the Landau-Zener mechanism in a trivial junction. In some sense, this remains a possibility for $fJ/2$ radiation in addition to Shapiro steps, quite apart from the circuit mechanism discussed here.

A final minor point: The authors say "Unfortunately, the present layout of our experiment does not allow for the implementation of TRS-breaking by application of a (local) magnetic field." This is confusing given that a magnetic field was applied - so a reference might help clarify.

In summary, this is an interesting manuscript that should certainly be published somewhere. However, because of the question of relevance to other claims of fractional JJ discussed above - I am not sure if this is appropriate for the broad readership of Nat. Comm.

Version 2:

Reviewer comments:

Reviewer #1

(Remarks to the Author)

For the second round of reviewing of this paper, the author have provided answers to the questions of the referees which has lead to a drastic enhancement of the supplementary material of the paper. The work of the author has also cleared a lot of the less clear points made in the original version of their work. I am a bit shocked however of the amount of change that was made to the supplementary which seems to indicate that the original submission was maybe made a bit too early, before a full supplementary material could be written. However, this issue has now been fixed. In summary I think that this version greatly enhances the quality of the work and the authors have satisfactorily answered my remarks and question regarding the first version of the work.

As I mentioned in my original review of this work I believe that the subject at hand and the research approach is of significant importance and as such, given the improvement made to the manuscript, I support the publication of the paper in Nature communications.

Reviewer #2

(Remarks to the Author)

The main question from my side was about overheating. I find the response unsatisfactory. The authors have performed simple estimates based on electron-phonon coupling and conduction to the leads. Both cooling mechanisms give too little cooling power to explain thermalization to 500 mK. In other words, they predict $T > 500$ mK. With some simple estimates I find that $T \sim 0.9$ K might naively be expected.

Setting the numbers aside, the authors are finding that heating should be a large effect, not a small one. One can easily imagine the estimates being off numerically, but simply arguing that heating is too big of an effect to explain the data hardly seems like a good reason to discard it. That said, I accept that the authors have analyzed the overdamped case and found consistency. This is also a minor point. I simply find the answer strange.

I find the current way that Laroche is cited in the introduction to be unclear. The phrasing "A 4π -periodic Josephson effect in photon emission has also been reported for InAs nanowire junctions [17], where an on-chip detector was used. The seemingly conflicting theoretical and experimental results of recent years motivate our renewed interest in the topic, extending the scope of our analysis to nonlinear dynamics effects." casts doubt on Laroche. Yet, the discussion with the Referee says that the authors do not expect period-doubling dynamics for Laroche. I am not sure if this is intentional, but a straightforward reading would leave things unclear as to whether this result draws Laroche into question.

Reviewer #3

(Remarks to the Author)

Dear Editor,

The authors of the manuscript titled "Period-doubling in the phase dynamics of a shunted HgTe quantum well Josephson junction" have revised the manuscript to include a discussion of similar fractional Josephson experiments in nanowires (Ref 17). With the addition of this reference, the manuscript now includes a discussion of their results provide guidance for experiments in other systems showing fractional Josephson junction peaks. The manuscript presents a relatively complete understanding of a likely mechanism for non-topological fractional Josephson peaks and how to avoid them. Therefore, I now believe that this manuscript is appropriate for the broad readership of Nat. Comm.

Response to the referees:

We would like to thank the reviewers for their careful reading of the manuscript and thoughtful comments. Following the suggestions, we have expanded the presentation of the DC characterization and now include a detailed discussion of I-V hysteresis and Shapiro step pattern in the Supplementary Information. We have also included a numerical simulation to demonstrate the mechanism by which missing Shapiro steps occur in the absence of an intrinsic 4π -periodic supercurrent component in the parameter range relevant to our experiment. Additionally, we have added numerical simulations to address white noise fluctuations, the role of the current-phase relation for the observability of the period-doubling effect, and the loading by the connection to the amplifier circuit. Finally, we present considerations how to improve the present experiment and meet the requirements for detecting the fractional AC Josephson effect. Here, we address the advantages of the on-chip detection method and raise awareness that the detector must also be included in the analysis.

Below we will address the concerns point-by-point:

REVIEWER COMMENTS

Reviewer #1 (Remarks to the Author):

This paper by Wei Liu et al. presents their work related to the microwave emission by a Josephson junction based on a HgTe heterostructure placed in an electromagnetic environment. The paper focuses on the different radiation lines that can be observed in this system in relation with how the different elements placed in parallel to the main junction influence the microwave emission by making new lines appear at half the Josephson frequency of the system. Because the system does not meet the necessary topological conditions, such doubling of the periodicity in the emission cannot have a topological origin. Therefore, the authors make the point that previous results obtained in such system should be re-examined in the framework of an emission modified by the electromagnetic environment of the junction.

I would like to start my review by saying that I believe that this type of paper is of a broad interest and valuable to the community as it brings alternative interpretation to sometimes sensationalistic papers. I find that this paper contains interesting results that might be worth being published in Nature Communications. However, I have several remarks and questions for the authors. In particular I find the paper hard to read, especially in relation to the notations used and the quantities measured that would gain from being explained more clearly. I would advise the author to extend the supplementary materials section so that the readers of the manuscript could have access to more complete data sets and analysis methods. In particular the data that have been renormalized are missing a dedicated section to elaborate on the normalization methods that were used. In the current state I cannot recommend this paper for publication in Nature Communications.

We have added section S3 in the Supplementary Information to address the issue of data normalization. Additionally, we added a paragraph describing the relevant quantities for circuit analysis earlier in the text.

Below I list my remarks that I hope the authors can answer.

1. In the introduction, the authors discuss TRS-breaking by a (local) magnetic field. At this point I have a question about the local field. Is it worth mentioning it at this point in the text? From what I understand, a small superconducting coil was used to generate the magnetic field. I would suggest, if this mention has no influence on the rest of the discussion, to get rid of this parenthesis.

We mention the magnetic field in the introduction of the paper to familiarize the reader with various scenarios required for observing 4π -periodic supercurrent. Breaking TRS is needed to restore 4π -periodicity in the system (cf. Fu and Kane, 2009). This is done by opening a gap in the dispersion of the TI weak link, resulting in gap openings in the dispersions of the ABS. It was suggested to break TRS by the exchange interaction of a ferromagnetic insulator in contact with the TI, or by a localized magnetic field in the TI material (e.g., the fringe field of a nanomagnet sitting next to the junction). The latter needs to be large enough to open a gap in the dispersion of the topological states. Although perhaps attractive to theorists, we believe the exchange coupling scenario is unlikely to be realized in experiments, and thus we only mention the second method.

We apply a very small magnetic field ($\sim 500 \mu\text{T}$) in order to modulate the phase in the junction and observe Josephson supercurrent interference. Later, we also use the method to “tune” the magnitude of the supercurrent in order to crossover between period-one dynamics and the period-doubling regime. Here, the fields are significantly smaller in magnitude than what is needed to open a gap in the dispersion of the TI weak link. Rough estimates yield $\sim 10 \text{ mT}$ for a gap of $\sim 1 \mu\text{eV}$ in the dispersion of the topological states. Breaking TRS is not our intention for applying the external field in the present experiment.

We have modified the wording to emphasize that a large and localized magnetic field is needed.

2. In page 4, the sentence “The numerical analysis suggest that the system is close to a period-doubling instability, causing the 4π periodic evolution of the junction phase ϕ .” Mentions a numerical analysis. This analysis is nowhere to be found in the manuscript. I would expect that it at least appears as supplementary material and be discussed more thoroughly.

This subsection is intended as a short introduction (or “preview”) for the main part of the manuscript. In particular, we refer to the RCLSJ circuit modeling that we present later in text. We slightly altered the wording to avoid confusion.

3. In figure 1.b, a scale bar is missing to indicate the size of the structure. At minima it should appear in the caption of the figure.

A scale bar does not work well for a view at an angle. We now added size information in the caption.

4. In figure 1.f&g arrows are presented to indicate the $n=1$ & $n=3$ missing steps. However, they are not mentioned in the caption. I believe it would make the figure clearer to add a mention in the caption.

We added a remark in the caption of the figure.

5. In fig.2, the schematic of the measurement circuit indicates a voltage V_{DC} (fig1.a) and another voltage V_S that seem to both connect to the junction/the shunting circuit. From what I understand, the voltage that is applied is V_{DC} that generates a current I_{DC} and V_S is the measured voltage down at low temperature that is used to extract the critical current. Similarly, V_J , if I understand correctly is the measured voltage across the Josephson junction that will be dependent on the shunting resistor and the rest of the circuit. Finally, in the text the notation \bar{V}_J is also used. What is the meaning of the averaging here? Generally, I think the author should do a better job at clearly defining the quantities they use in their work in order to avoid any confusion on the reader's side.

The quantities V_S , V_J , I_S , and I_J have DC and AC components. We use barred symbols when we refer to the DC average component of a quantity.

V_S is the voltage drop across the parallel circuit of sample and biasing resistor. For detection, we split the signal into its DC component \bar{V}_S and the microwave signal using a bias tee. V_J is the instantaneous voltage across the junction. We cannot measure its DC component, \bar{V}_J , directly, but it is inferred once the wiring resistances have been extracted. \bar{V}_J is used to determine the Josephson frequency and plot the I-V characteristics of the loaded (by the "external" circuit elements R_W , R_S and L_S) junction in Fig. 3. Here, \bar{I}_J refers to the DC current component that flows in the junction branch of the circuit.

We added a description of Fig. 2a&c earlier in the text to prevent confusion and modified the labels in Fig. 2a for consistency with symbol naming conventions.

6. In page 6, the page begins with "At $B=0$, the current-biased device exhibits the hysteretic current-voltage (I-V) characteristic of an undershunted Josephson junction. I have two comments in this instance. The first one may be personal due to a gap in my knowledge but I don't understand the term "undershunted". I understand what the shunted part comes from but "undershunted" seems to come in opposition to "overshunted" or "normally-shunted". What is the criterion for being undershunted? Also looking at refs. 33 & 34 I couldn't find this expression used either. Could the authors clarify what they mean by "undershunted"? My second point is related to the $B=0$ statement. Could the author present (probably best in the supplementary) curves at $B \neq 0$? How does the hysteresis evolve at finite field?

This is a typo. It is supposed to read "underdamped." An underdamped junction has a Stewart-McCumber parameter $\beta_c > 1$, as opposed to a critically-damped ($\beta_c \approx 1$) or overdamped device ($\beta_c < 1$). The latter two cases show no hysteresis in the I-V characteristic. We fixed the typo in the resubmission. For $B \neq 0$, the hysteresis diminishes as I_c decreases. It vanishes when $\beta_c < 1$. We discuss I-V hysteresis extensively in Section S1.1 of the Supplementary Information.

7. Later in the same paragraph, the main claim of the paper appears as the first and third steps of the Shapiro steps are missing in fig.1.f, thus mimicking what is expected for a fractional system.

Our main message is that the Josephson emission measurements, as performed using the method of Deacon et al., Phys. Rev. X 7, 021011 (2017), exhibit period-doubling dynamics if the parasitic inductance in the circuit is too large. A large parasitic inductance can also affect Shapiro step measurements. This has been discussed in the early literature on Josephson devices when weak links were often realized by macroscopic structures. However, the problem of parasitic inductance is often forgotten when micro-fabricated Josephson devices of novel materials are investigated.

As we point out, several mechanisms for missing Shapiro steps have been considered in literature. Following the analysis in the manuscript, it becomes clear that an inductive reactance in the circuit enables 4π -periodic phase dynamics in our experiments. In the comparison experiment with significantly reduced shunt inductance, the hallmarks of 4π -periodic phase dynamics (missing Shapiro steps and emission at $f_J/2$) are absent. This means that an intrinsic 4π -periodic supercurrent component is too small to be detected under the experimental conditions. Vice versa we demonstrate in Section S1 of the revised Supplementary Information how patterns with missing odd Shapiro steps may occur when a realistic scenario for the junction environment is considered.

This is indeed true at low power, but I can see traces of these steps at higher power. Could the authors explain the role of the power and compare to what is expected for a fractional system? Also, from this data I imagine that the authors have at their disposal dI/dV maps as a function of voltage and power. I think it would be helpful to include some of those maps in the supplementary to have a complementary way to look at the data and maybe even make some comments on the comparison with the expected behavior in topological systems.

For a comprehensive answer to this question, we kindly direct the reviewer to Park et al., Phys Rev B 103, 235428 (2021), who calculate the effect of a 4π -periodic supercurrent component on the Shapiro steps in the framework of the RCSJ model in the experimentally relevant limits. Let us summarize the main points: At low drive frequencies the phase of the junction cannot lock to the drive for Shapiro step voltages that correspond to odd multiples of the drive frequencies. The effect is most easily observed in the low power regime, where the Shapiro step amplitude “ramps up” (for current-biased measurements). Generally, it depends strongly on microwave frequency, shunt-capacitance, and relative size of the 2π - and 4π -supercurrent components whether Shapiro steps are suppressed. Only for a large fraction of 4π -periodic supercurrent, Shapiro steps are suppressed in the high power regime. For smaller fractions of 4π -periodic supercurrent, large values of β_C result in partial suppression of odd-numbered Shapiro steps. In this case, lower capacitances yield smaller frequency ranges for which Shapiro steps are suppressed. In the context of the fractional Josephson effect by means of “Majorana bound states,” the intrinsic dynamics of the “Majorana qubit” needs also be considered [see, e.g., Wang et al., Phys. Rev. Lett. 129, 257001 (2022)]. Here, too, suppression of the first (and other) Shapiro step(s) is expected, but the visibility is related to the parity relaxation processes of the “Majorana qubit state” and thus the observability depends on a number of additional parameters.

The data in Fig. 2f are thus reminiscent of a system with a small fraction of 4π -periodic supercurrent and large β_C . As mentioned above, the experiments in circuit configuration C2 suggest that an intrinsic 4π -

periodic supercurrent component has to be very small if present. However, even at the lowest gate voltages, for which edge channel transport is expected to dominate the signal, we cannot detect emission at $f_J/2$. Therefore an alternative explanation for Shapiro step suppression is more plausible.

Period-doubling dynamics by parasitic inductance offers another mechanism for Shapiro step suppression. It was first pointed out by D. B. Sullivan, et al., *Generation of Harmonics and Subharmonics of the Josephson Oscillation*, Journal of Applied Physics 41, 4865 (1970), that an inductive shunt leads to missing Shapiro steps as phase-locking is inhibited. In our case, the junction is shunted by a LCR-resonant circuit formed by the wirebond inductance L , the capacitance C in the flatband cable that connects to the sample holder, and the series resistance R_W in the wiring layer. For reasonable estimates of the values L and C , and frequencies in the low GHz range, the reactance of the shunt is inductive and Shapiro steps are suppressed. We have verified the basic mechanism in RCLSJ simulations. However, we have not attempted to produce exact matches with experimental Shapiro step patterns for a large range of microwave frequencies.

The combined experimental observations point at period-doubling due to a parasitic shunt inductance as most plausible explanation for the 4π -periodic phase dynamics in our device.

We added complementary plots of the Shapiro data in Fig. 1 in Section S5 of the revised Supplementary Information as requested. We also added an example of a Shapiro step calculation with parameters reasonably close to the circuit in our experiment. It exhibits suppression of the first and third Shapiro step.

8. Page 6 again, it is written that the fridge has a base temperature of 7 mK. However, earlier in the results section it is mentioned that the base temperature is 35 mK. Please clarify this point.

We used two refrigerators for our measurements. DC measurements were carried out in a refrigerator with specialized DC wiring (LC-, RC-, and copper powder filters) and a base temperature of ~ 35 mK. The emission measurements (and the Shapiro step measurements with on-chip shunt) were carried out in a refrigerator with RF wiring and a base temperature of 7 mK.

9. In fig. 3.g I see observe another resonance that appears in the simulation that is not observed in the experimental data. Does it correspond to $f_J/3$? $f_J/4$? Could the authors comment on this?

Indeed, there is a feature at $f_J/3$ in the simulation. In the experimental data, it is washed out by the broad resonant feature centered around junction voltage $\frac{hf_{LC}}{e}$. We observe the loss of contrast also in the simulations in Section S4.1 where we broaden the resonant feature by a larger amount of white noise fluctuations in the circuit.

The mechanism for $f_J/3$ emission is identical to emission at $f_J/2$. For large enough capacitance and high bias, 6π -periodic phase evolution becomes accessible in RCLSJ devices, cf. Neumann and Pikovsky, The European Physical Journal B - Condensed Matter 34, 293 (2003). We did not observe a pronounced feature at $f_J/3$ in the measurements taken on this device, yet we already had observed this phenomenon in a previous experiment on a similar device without gate, where the $f_J/3$ -line features

prominently. We present the data of this measurement in the Supplementary Information of the revised manuscript. We also discuss the $f_j/3$ line in the main text of the revised manuscript.

10. Concerning the simulations in figure 3, it is said in the text that finite temperature is simulated by adding white noise current sources in parallel to the resistors (p9). My naïve guess would be that this has the effect of broadening the resonances such as those seen in fig3. My question is related to the amplitude of this white noise. My feeling is that it is underestimated as the width of the resonance in simulations in fig. 3 is very small compared to the experimental data. Are there other mechanisms that could explain this additional broadening. What does the simulation look like for various values of the noise amplitude?

In Fig. 3, we introduced only a small amount of white noise. This is done to allow for better visibility of the emission lines in the simulations. It is not meant to reflect a physical measure of the noise temperature of the sample. This is made clear in the text of the revised manuscript.

We present additional numerical simulation for various amounts of white noise fluctuations in Section S4.1 of the revised SI. The linewidth of emission and the width of the resonant features increases with increasing white noise. However, in the simulations only thermal white noise of the resistors in the RCLSJ model is considered, whereas the shot noise component $\langle i^2 \rangle = 2 q_{\text{eff}}(V) I_{\text{qp}}$ that arises from the quasiparticle current I_{qp} , is omitted as we have no direct way of inferring the effective charge q_{eff} from the data. Let us keep in mind that there is no contradiction with the fact that the emission linewidths do not broaden with increasing \bar{V}_J as the DC component of the junction current with \bar{I}_J stays approximately constant in the measured ranges [cf. I-Vs in Fig. 3], and the effective charge $q_{\text{eff}} = q_{\text{eff}}(\bar{V}_J)$ transported by multiple Andreev reflections decreases with increasing voltage.

Let us also point out that the environment (coupling to cavity resonances and biasing/detection circuit) will contribute to noise fluctuations and broadening of the emission linewidths [cf. Hofheinz et al., Phys. Rev. Lett. 106, 217005 (2011), Laroche et al., Nature Communications 10, 1 (2019)] but is hard to quantify in terms of its frequency dependence and noise temperature in our measurement configuration. Additionally, we need to consider that real circuit elements have loss, and the quality factor of the LC-resonance in our simplified circuit is likely lower compared to the simulations. This broadens the LC-resonance and affects the phase dynamics.

11. In fig. 4 I find it hard to understand what is plotted in figures b, e and h. From what I understand figure a, d and g correspond to the emission as a function of \bar{V}_J and the gate voltage (hence the density) and the z axis correspond to the power detected at the frequency f_d (which is not specified). I would tend to think that $f_d = 4.54$ GHz as it is the same frequency used in b, e and h. However, I am failing to understand the link between the first and second row figures. If b, e and h are simply cuts from a, d and g along the dotted red line I don't understand the data that is plotted as the color code coded data in the map doesn't have the same evolution as a function of the various parameters as the b, e and h lines. Does it have to do with normalization procedure? Then it should be explained more clearly. Could the author help me understand better the quantities plotted in these figures? What do the author mean by "the integrated power of the emission peak" precisely. I find it confusing and that this harms

the readability of the paper. In short, could the author make it clearer what is the relationship between P_{f_j} and the spectral density PSD_d ?

The figures in rows 1 and 2 of Fig. 4 serve different purposes. Fig. 4a,d,g show the measured power spectral density (PSD_d) normalized by the maximum amplitude of PSD_d for each value of V_g , B , T , respectively. This allows us to track the junction voltages for which strong microwave signals are detected at detector frequency $f_d=4.54$ GHz. Fig. 4b,e,h show approximate measures of the emission power P_{f_j} and $P_{f_j/2}$, associated with period-one and period-doubling dynamics. To this end, we have adopted a suitable convention [cp. Haller et al., Phys. Rev. Res. 4, 013198 (2022)] by integrating PSD_d over the respective lineshape along the \bar{V}_J -axis.

We mention the detector frequency for panels a, d, g in the figure caption and rephrased the discussion of the figure in the revised manuscript.

12. In figure 4.e the authors plot the field dependence of the critical current on top of the integrated power and compare these two quantities, in particular insisting on the similarity of the shape of the P_{f_j} curve at large field that follows that of the critical current. Isn't there a claim that could be made here that the sum of the emission at P_{f_j} and $P_{f_j/2}$ reproduce somehow the whole field dependence of the critical current or at least they are link in some way. I also find the explanation of the link between $P_{f_j/2}$ and I_c to be very qualitative but missing a bit of physics to get a sense of why these two quantities are linked and the underlying phenomena.

For the case of a single emitted frequency and ideal voltage-biasing conditions, the Josephson current oscillates with $I(t) = I_c \sin \frac{2eV_J}{\hbar} t$, thus the signal power P for fixed voltage $V_J = \text{const}$ scales as $P \propto I_c \times V_J$. Therefore, when detecting emission for a fixed frequency, f_d , the power scales with $I_c(B, V_g, T)$ as shown in Fig. 6.

For the plots in Fig. 4, the situation is vastly more complicated, as the time-evolution of $V_J(t)$ and $I_J(t)$ result from the slow-fast phase dynamics described in Neumann and Pikovsky, The European Physical Journal B - Condensed Matter 34, 293 (2003), and the harmonic content distributes between additional frequencies besides f_j and $f_j/2$. Thus a simple (or weighted) addition of P_{f_j} and $P_{f_j/2}$ is not expected to reproduce the shape of I_c in plots such as Fig. 4 or 5, but the specifics of the dynamics (including the voltage-dependent dynamic resistance) must be taken into consideration for each parameter value V_g , B or T . We therefore focus on a qualitative description of the simplest case and demonstrate the dependence of the crossover on the ratio of Josephson to shunting inductance in Fig. 5.

13. In page 10 the second to last paragraph starts with "Fig. 3c depicts a 2D map of the Fourier transform...". This data is a simulation; however, I find that it is not apparent from the beginning. I would advise the author to make it clear when they are presenting a simulation vs. actual measured data.

We added a sentence to make this point clear to the reader.

14. Page 11 the second line mentions “a wide range of theoretical work, analog simulation, and computer numerics” but no citation accompanying this sentence. It would be worth citing some of these works or reviews.

We now refer the reader to Ref. 16-22.

15. Page 12, in the sentence “It follows a Fraunhofer-like diffraction pattern” referring to fig.4f, I agree with the statement. However, this behavior is better seen in fig. 6 e & f. I would advice this figure to be mentioned here as well or maybe that the Fraunhofer discussion comes a bit later so that fig 6 had already been introduced.

Here, we refer to the critical current I_c that follows a Fraunhofer-like diffraction pattern. This is to show that we, indeed, modulate the supercurrent in the junction and serves for comparison with panels c and i. We do not discuss the emission signal. We believe it would be confusing for the reader to make a reference to a figure that shows emission data at this point in the text and therefore we respectfully decided to not follow this suggestion.

16. Page 13, the first full paragraph concludes with a comment on the effect of the temperature increase on the features of the P_j . I guess this means that the authors have this data and I think they should consider adding it as a section to the supplementary materials.

We added traces for 7 mK in Fig. 5 (open symbols) and mark α_c as well as the transition region in the plot.

17. In figure 6, it seems to me that there is a small spectral weight that is barely visible in the PSD maps. Maybe the authors should discuss that at the end of page 13 as they explicitly write “we observe microwave emission only at the Josephson frequency (f_J)”.

We point out a small $2f_j$ emission signal in the revised manuscript.

18. In figure 7, the caption of the figure reads “the data are normalized for best visibility”. At this point my feeling is that more details about the normalization procedure are needed. Is the normalization global or at constant power or constant voltage?

We added information about the normalization procedure in the caption of the Fig. 7.

19. Page 16, the first paragraph concludes with “We have carried out additional numerical simulations for realistic circuit-loading conditions, finding no qualitative changes in the dynamics”. Where are those numerical simulations? Shouldn't they appear in the supplementary to support your claim?

We present a number of additional numerical simulations regarding white noise fluctuations, current-phase relation and circuit-loading by the detector in Section S4 of the revised Supplementary Information.

Response to request for comment on report of Reviewer #3

1 - From what I understand from the Nature communication by Laroche et al., the authors study the 4π Josephson effect in InAs nanowires with a wire coupled to a broad band detector. For me the paper under review and that paper address different aspects of the problem. The Nature communication follows the path trying to identify clear signatures of the 4π Josephson effect while taking into account the electromagnetic environment of their Junction. However, this paper lacks full time simulations that would allow to compute the expected shape of the $I(V)$ characteristics and study the potential emergence of non-topological 4π periodicity. In contrast the paper under review examines the case of a system NOT in a topological configuration that nevertheless shows signs of 4π periodicity. Therefore the paper is trying to prove that the simple disappearance of Shapiro steps cannot be considered as proof of a topological behavior of their system. I agree with the referee that a comment of the author about this work and at the very least a citation would be necessary in order to draw a complete picture of the recent experiments on the subject. Therefore I think that this paper brings enough new information compared to the Nat. Comm. paper to guarantee publication (as the referee says, whether or not this should be in Nature communications remains to be seen after the authors answer the first round of comments from the referees).

2 - Relating to the reference [19] of the paper under review. I agree with the referee that the principle of the two papers are quite similar. However, the 1998 paper does not study the 4π periodic Shapiro steps per say. It does not exclude its appearance explicitly but fails to comment on that. For this reason I think the Liu paper pushes further the ideas of [19] applying them to a very specific situation. Therefore I don't think it would be so straightforward to get to the result of the submitted paper solely based on [19].

3 - Once again in my understanding of the Laroche paper, the scope of the work is quite different despite the thematic overlap and the take-home message of both papers are quite different. As I said in my first answer I think a decision regarding the adequacy of Nat. Comm. for publication remains to be decided after the authors answer the questions of the referees.

Reviewer #2 (Remarks to the Author):

In “Period-doubling in the phase dynamics of a shunted HgTe quantum well Josephson junction” Liu et al. study Shapiro steps and microwave emission from a HgTe Josephson junction. They observe a missing Shapiro step at low magnetic field, similar to [3,4]. They observe subharmonic ($fJ/2$) Josephson radiation at low magnetic field, low temperature, and not-too-negative gate voltage. Their findings are explained by the nonlinear dynamics of a well-motivated circuit model. Moving to an on-chip shunt resistor removes the $fJ/2$ radiation and restores all Shapiro steps, consistent with the model.

There have been many works associating missing Shapiro steps with topological phases. It is generally understood that there are several non-topological reasons that these can occur, as the authors discuss in their introduction.

This work is to my knowledge the first to deeply understand the origin of the missing step, and to be able to restore the expected “textbook” behavior. It therefore plays the role of clearing up questions that linger in the old literature, and presenting a potential path forward for using Shapiro steps and Josephson radiation as a meaningful diagnostic of topological superconductivity.

This contribution is significant enough to appear in Nature Communications.

I do not have major scientific concerns about the work. A few minor questions that I would like the authors to address:

(1) The authors identify hysteresis in their current-voltage characteristics as being from underdamped phase dynamics. I am skeptical that this identification can be made. Have the authors critically assessed that this could be from overheating? A simple estimate would be to estimate the effective equilibrium temperature T^* in the normal state accounting for Joule heating, and compare the observed “re-trapping” current with $I_c(T^*)$.

We present additional data on the I-V hysteresis in Section S1 of the revised SI.

The hysteresis is compatible with the dynamics of an underdamped junction with parasitic capacitance of 0.28 pF over the entire range of gate voltages, V_g [Fig. S1]. A small deviation is visible at large values of V_g . We attribute this deviation to the presence of a circuit resonance.

We show temperature-dependent switching and retrapping current data for $V_g = 0.5$ V [Fig. S2a]. Unfortunately, this is the only dataset of this kind available as we did not pay much attention to I-V hysteresis in the initial DC characterization of the device. We observe the onset of hysteresis at critical damping, i.e., when the Stewart-McCumber parameter $\beta_c \approx 1$. This supports our interpretation of the hysteresis originating from underdamped phase dynamics. The deviations in retrapping current (as compared to the simple RCSJ model in Fig. S1) occur in voltage regions where circuit resonances affect the dynamics of the Josephson junction. We demonstrate this by plotting the retrapping branch of

the I-V curves for a narrow range of temperatures ≥ 500 mK. For the larger temperatures, two switching events are observed. The second one occurs between the two voltage values that are associated with the known circuit resonance in our setup [cf. Fig. 2b]. Let us point out that these resonances are resonances in the DC measurement setup and not identical with the resonance at f_{LC} in circuit C1. Further, this is not a singular observation, but we have tracked these features in a series of previous measurements taken in this setup since becoming aware of the issue. We also show a representative dataset for switching and retrapping currents as a function of magnetic field (i.e., a “Fraunhofer-like pattern”). Again, I-V hysteresis is observed when $\beta_c > 1$, and the magnitude of the retrapping current is modeled well by the approximation formula.

In conclusion, the I-V hysteresis in the experimental data is satisfactorily described by conventional RCSJ-like dynamics of an underdamped Josephson junction. We do not require “overheating” of the electrons to describe the observations.

We close by summarizing our understanding of the “overheating” concept in mesoscopic SNS junctions and discuss attempts to model the effective temperature:

I-V hysteresis in mesoscopic SNS junctions has frequently been attributed to out-of-equilibrium quasiparticle populations with an “effective equilibrium temperature T^* ”. The concept was introduced by Courtois et al., Phys. Rev. Lett. 101, 067002 (2008) in the context of SNS junctions with metallic wire weak links. The presence of nonequilibrium quasiparticle populations is caused by Joule heating, and the effective electron temperature was found to be limited by the thermal conductance of the superconducting electrodes of the SNS junction.

Later, the idea of overheating was adopted by Le Calvez and co-workers, Communications Physics 2 (1), 4 (2019), to explain I-V hysteresis and the observability of the “missing Shapiro step phenomenon” in topological insulator Josephson junctions. The authors introduced a “thermal RSJ” model and identified electron-phonon (e-ph) interaction as the leading mechanism of heat dissipation. We remain skeptical of this work for two reasons:

Firstly, it appears that the device capacitance is grossly underestimated by using a parallel plate capacitor model. This unfortunate mistake has been made by our group in previous work, too. However, it would appear desirable to establish the correct device capacitance value and compare hysteresis and thermal hysteresis for a model that includes the capacitive shunt.

Secondly, the authors support their claim of e-ph scattering in the topological insulator being the leading heat dissipation mechanism by fitting the hysteretic I-V in Supplementary Figure 7 of this work. It yields a sensible result under the condition that the e-ph interaction can be modeled analogous to a dirty metal film [Echternach et al., Phys. Rev. B 46, 10339-10344 (1992)] which seems hardly plausible for the TI.

To estimate heat dissipation by e-ph scattering in our sample, we take values for the e-ph coupling from experiments on HgTe quantum well structures [S.U. Piatrusha, et al., Phys. Rev. B 96, 245417 (2017)]. This yields heat dissipation of <1 pW at 500 mK assuming our junction geometry and the experimental parameters in Fig. S2a. This is opposite to Joule heating of ~ 7.3 pW.

We also estimate heat transport via the leads. For this, we take into account four leads (with a typical distance between TI interface and normal metal reservoir/bonding pads $\approx 70 \mu\text{m}$) with a normal state lead resistance $\approx 14 \Omega$ and a superconducting gap $\Delta \sim 1 \text{ meV}$. Here, we infer the resistance based on measurements of the geometry of the electrodes and the typical sheet resistance of our sputtered Nb films. Both are determined in separate experiments. Applying eq. 2 of Courtois, et al., then yields $7 \times 10^{-19} \text{ W}$ for $T^* = 500 \text{ mK}$.

In both cases, the dissipated heat is too small by one or several order(s) of magnitude, and a more comprehensive approach to modeling heat transport and dissipation is needed. It likely has to include a self-consistent calculation of the superconducting gap [i.e., “inverse proximity effect” $\Delta = \Delta(I, T^*)$], and the excess current contribution (Andreev reflection at the interfaces) has to be properly accounted for in the model. [Dane et al., *Self-heating hotspots in superconducting nanowires cooled by phonon black-body radiation*, Nat Commun 13, 5429 (2022); Ibabe, et al., *Joule spectroscopy of hybrid superconductor–semiconductor nanodevices*, Nat Commun 14, 2873 (2023).] At this point in time, we have not engaged in such modeling, which requires establishing accurate values for a set of additional model parameters. However, we feel confident in saying that it is not a “simple estimate” to determine a meaningful value for T^* as expressed by the referee.

(2) Can the authors supply circuit parameters for their on-chip resistor solution? I am asking for circuit parameters including stray inductance/capacitance, not merely the nominal lumped-element resistance. Could this be a practical way of realizing a high impedance environment for dual Shapiro physics and the like?

We did high-frequency EM modeling for this circuit as well. For the simulation, we defined a shunt resistance of 10Ω between the two contacting pads. This yields an equivalent inductance $L \approx 0.5 \text{ nH}$ in the relevant frequency range of 1-7 GHz.

Regarding the second question, we doubt that this method is a ‘practical’ solution as it requires macroscopic contacting pads for connecting the resistor. The wiring layer will contribute parallel capacitance and shunt at high frequencies. To provide a size reference: The smallest package for the HF resistors, we used in the manuscript, has contact pads $\sim 120 \mu\text{m} \times 390 \mu\text{m}$. For the low resistance values used here, the parasitic capacitances and inductances in the surface-mount resistor lead to deviations from the ideal behavior starting in the range of 10 GHz. (Ref: Vishay datasheet). This may further restrict applicability.

Response to request for comment on report of Reviewer #3

I do not think the mere existence of Laroche et al. is a major challenge to the novelty or significance of the present work. However, I am quite surprised that it was not cited and discussed in a clear way.

Referee 3 has done a service by pointing this out. It would be interesting to hear why the authors did not cite this work.

I think Ref. 19 is cited appropriately. A reasonably skilled expert would certainly take away from Ref. 19 that dynamics are complex and need to be thought about carefully. Certainly this has been appreciated. But knowing that some previous $fJ/2$ observations were due to period-doubling dynamics, which can be completely suppressed with an on-chip resistor, is not possible just from reading [19].

Properly citing and discussing Laroche et al. will most likely alter this work, but in my opinion it is unlikely to compromise novelty and impact.

Citing and discussing Laroche et al. will cause the authors to deal directly with the fact that prior works have indeed considered junction dynamics more carefully. On the other hand, I am not sure to what extent Laroche et al. really engineered their environment, as opposed to fitting for effective environmental RC values that give overdamping (and are consistent with earlier work pioneering their technique). I do find it questionable if the RC parameters Laroche et al. determine are really correct. It seems plausible that the capacitance of their Al/AlO_x test junction is part of their RC fit, but that this is essentially absent in their real InAs device. This would lead to a very different environmental impedance than what is modeled, possibly altering the phase dynamics. This makes it all the more valuable to see that the environment can be directly controlled with an on-chip resistor, and that this is definitely needed in some approaches.

Reviewer #3 (Remarks to the Author):

Dear Editor,

The manuscript titled "Period-doubling in the phase dynamics of a shunted HgTe quantum well Josephson junction" compares extensive RF measurements of emitted radiation from a HgTe Josephson junction (JJ) with careful microwave simulations to show that the $fJ/2$ frequency radiation peak often associated with topological Josephson junctions, notably in 3 previous papers (Refs. 3,4 and 14) by the present group can be attributed to non-linear dynamics from circuit resonances similar to those studied in trivial JJs in Ref 19. Apart from several theoretical works cited in this manuscript - related claims have been made for the doubled Shapiro step (Refs 11 and 12), which was motivated by Ref 1 in the nanowire. The present manuscript seems to be the first to compare fractional Josephson radiation from a possibly topological JJ to simulations of such radiation from a trivial JJ (à la Ref 19) and show that this

is trivial. Thus, this work apart from shedding light on previous experiments from this group i.e. Refs 3,4 and 14 can potentially provide guidance for the field of topological JJs - though the phenomenon reported here is essentially the physics reported in 1998 (i.e. Ref 19).

While I agree with the findings of this manuscript and consider them significant enough to be published whether they should be published in a journal for a broad readership depends on the level of guidance this manuscript can provide to the field of topological JJs beyond Refs. 3, 4 and 14. Radiation measurements, though preferable are difficult and therefore there are not a lot of experiments in this area. In this regard it is notable that Nature Communications volume 10, Article number: 245 (2019), which claimed to observe fractional JJ radiation in nanowires, is not cited in the present manuscript.

In fact, this paper claims "This on-chip detector²⁹, coupled via capacitors CC to the NW junction (see Fig. 1b for the schematics and Fig. 1e for an optical image of the device), was engineered to result in an overdamped microwave environment characterized by a single $f_c = (2\pi RC)^{-1} \approx 28$ GHz cutoff frequency with $R = 538 \Omega$ and $C = 10.4$ fF (see Supplementary Fig. 2). The resulting broadband coupling to the detector⁷ inhibited higher order photon emission, which could mimic the 4π -periodic Josephson effect^[30]." where Ref 30 is Cawthorne et al (Ref 19 in this work).

Specifically, they claim to have ruled out the effect discussed in this work. I think it is crucial for the present manuscript to discuss this claim to some degree, which is of course limited given that the authors may not have access to the relevant data from this paper to determine the validity of the claim.

Laroche, et al., present a different measurement concept, material system and physical effect than we analyze in this manuscript. We agree with the authors in that we do not expect the period-doubling mechanism to occur in their devices, based on the information published in the paper. Therefore, we did not think to include the reference in our original draft. In our revised manuscript, we follow the referees' suggestion and broaden the scope of the discussion to include additional guidance to the experimentalists who are interested in detecting the fractional Josephson effect. We thus elaborate on possible improvements to the measurement configuration and the pros and cons of the on-chip detection scheme as realized in the work cited by the referee.

For a device in this wire geometry and with the published parameters, the paper's claim regarding the frequency dependence of the environment is plausible. The role of the environment is discussed in detail in a previous publication of the group [van Woerkom, et al., Josephson radiation and shot noise of a semiconductor nanowire junction, Phys. Rev. B 96, 094508 (2017)]. Motivated by the referee, we simulated the nanowire and environment with and without coupling to a RCLSJ model for the SQUID detector at half flux bias [Fig. 1b of Laroche et al., Nature Communications 10, 1 (2019) and parameters taken from the Supplementary Information]. The nanowire was modeled with a pure $\sin \varphi$ current-phase relationship. As expected, we did not find that the inclusion of the detector branch (i.e., inductance of the SQUID) changed the behavior and found no evidence for the period-doubling effect, which is the topic of our present manuscript.

The present manuscript suggests: "We stress that the very simple modeling, we performed above, is sufficient to explain all salient features of the emission experiment". However, it is not clear that this

would be applicable to many topologically trivial JJs. For example it is unclear if such modeling is feasible when the current-phase relation is more complicated which is for example the case even with many topological materials Phys. Rev. Lett. 114, 066801 – Published 9 February 2015 and Phys. Rev. B 100, 064523 (2019). Do the authors suggest determining the current-phase relation in some independent way? This would be good for the manuscript to discuss.

The wording of the question is a bit ambiguous. We hope to provide useful information by stating the following points:

Firstly, the EM modeling of the circuit and RC(L)SJ simulations of the junction dynamics can be carried out for any current-phase relation (CPR). Clearly, the current-phase relation has an impact on the dynamics, and half-frequency emission and missing Shapiro steps will not be observed for certain CPRs and shunting conditions.

In the context of 2π - and intrinsic 4π -supercurrent components with different shunting capacitance, the phase dynamics has been explored [e.g., in Park et al., Phys Rev B 103, 235428 (2021)]. Here, the suppression of Shapiro steps depends on the relative magnitudes of the amplitudes, the frequency and the Stewart-McCumber parameter. It is our understanding that the period-doubling dynamics adds to the phenomenology, and can also be observed for trivial Josephson junctions. This does not mean that all trivial Josephson junctions will feature period-doubling dynamics. We show in Section S4 of the revised supplementary materials that a large $\sin 2\phi$ component of the CPR stirs the phase dynamics in favor of 2ϕ periodicity, as evidenced by the large emission amplitude at $2f_J$. A separate measurement of the CPR is advisable.

Although we have not measured the CPR of the devices in this experiment, we carried out a study of the CPR in similar Josephson devices recently [W. Liu, et al., *Phase-dependent supercurrent and microwave dissipation of HgTe quantum well Josephson junctions*, arXiv: 24112.17550 [cond-mat], 2024]. In these experiments, we coupled an RF SQUID, into which the topological Josephson junction is embedded, to a coplanar-waveguide (CPW) resonator for read-out. The CPR can be extracted from the complex admittance in a straightforward way. The junctions exhibited moderately skewed CPRs with an amplitude ratio of the $\sin 2\phi$ to $\sin \phi$ components between 12% and 25%. These values are similar to earlier measurements by the Moler group that are quoted by the referee.

A minor point is that the discussion of circuit details at the bottom of page 7 seems to present details that are difficult to follow and not obviously relevant for a general reader. Maybe the authors should consider moving this discussion to a different section i.e. methods or Supplement.

We moved part of the technical information to the Methods section of the revised manuscript. At the same time, we extended the discussion of the equivalent circuit, following the request of Reviewer 1 to explain the quantities in our plots more clearly.

Another relatively minor point, in the context of the discussion of radiation "detection of microwave photons emitted by the voltage-biased Josephson junction" is that Ref 10 explicitly calculates the $fJ/2$ radiation peaks from the Landau-Zener mechanism in a trivial junction. In some sense, this remains a

possibility for $f_J/2$ radiation in addition to Shapiro steps, quite apart from the circuit mechanism discussed here.

Midgap Andreev bound states and Landau-Zener transitions (LZT) are both plausible mechanisms for “intrinsic” 4π -supercurrent in topological insulator junctions as we point out in the introduction of the manuscript. In the present context, however, we are confident in ruling out both mechanisms. On the one hand, the frequency dependence in the measurements with large parasitic inductance fits very well to the period-doubling dynamics mechanism but not LZT activation. On the other hand, the emission pattern shows no $f_J/2$ (or $f_J/3$) emission when the shunt inductance is reduced in the on-chip shunted measurement configuration, and the Shapiro step pattern becomes trivial. Taken together, the three observations contradict midgap Andreev bound states or LZT as mechanism for the 4π -periodic phase evolution.

A final minor point: The authors say "Unfortunately, the present layout of our experiment does not allow for the implementation of TRS-breaking by application of a (local) magnetic field." This is confusing given that a magnetic field was applied - so a reference might help clarify.

We applied a magnetic field much smaller than what is needed to open a gap in the dispersion. This is now addressed in the manuscript.

In summary, this is an interesting manuscript that should certainly be published somewhere. However, because of the question of relevance to other claims of fractional JJ discussed above - I am not sure if this is appropriate for the broad readership of Nat. Comm.

We thank the referee for the thoughtful comments and hope, we addressed all concerns adequately.

On behalf of the authors,
With best regards,

Martin Stehno

Response to referees:

We thank the referees again for their careful reading of the revised manuscript. Below we address remaining concerns and summarize changes made to the text.

REVIEWERS' COMMENTS

Reviewer #1 (Remarks to the Author):

For the second round of reviewing of this paper, the author have provided answers to the questions of the referees which has lead to a drastic enhancement of the supplementary material of the paper. The work of the author has also cleared a lot of the less clear points made in the original version of their work. I am a bit shocked however of the amount of change that was made to the supplementary which seems to indicate that the original submission was maybe made a bit too early, before a full supplementary material could be written. However, this issue has now been fixed. In summary I think that this version greatly enhances the quality of the work and the authors have satisfactorily answered my remarks and question regarding the first version of the work.

As I mentioned in my original review of this work I believe that the subject at hand and the research approach is of significant importance and as such, given the improvement made to the manuscript, I support the publication of the paper in Nature communications.

Reviewer #2 (Remarks to the Author):

The main question from my side was about overheating. I find the response unsatisfactory. The authors have performed simple estimates based on electron-phonon coupling and conduction to the leads. Both cooling mechanisms give too little cooling power to explain thermalization to 500 mK. In other words, they predict $T > 500$ mK. With some simple estimates I find that $T \sim 0.9$ K might naively be expected.

Setting the numbers aside, the authors are finding that heating should be a large effect, not a small one. One can easily imagine the estimates being off numerically, but simply arguing that heating is too big of an effect to explain the data hardly seems like a good reason to discard it. That said, I accept that the authors have analyzed the overdamped case and found consistency. This is also a minor point. I simply find the answer strange.

We are sorry that the referee took our response so badly and apologize for the irritation that we caused. Allow us to assure, we did not intend to “brush off” the remarks---or take the concerns of the referee lightly. Up until the present study, we considered electron heating as the likely origin of hysteresis in our topological insulator Josephson junction. In the course of the research that led to this manuscript, we realized the period-doubling effect arises from a variety of “underdamped” junction dynamics in the presence of large spurious inductance in the shunting branch of the circuit. This realization forced us to reevaluate the mechanism of I-V hysteresis in our devices. In the Supplementary Information of the revised manuscript, we added a detailed discussion of hysteresis due to underdamped dynamics [Section S1.1] to document that link. We agree with the referee that the electron temperature must be elevated to some degree. However, the simulations in Fig. S1, the analysis of Figs. S2 and S3, and the observation of period-doubling effect in the shunted circuit C1

directly point at underdamped phase dynamics being the key ingredient here. Electron heating does not dominate.

In regards to heat dissipation in Fig. S2, let us point out that our estimates are based on the proposed models in the literature (Courtois et al., 2008 and Le Calvez et al., 2019). These yield an “effective” temperature in the single-digit Kelvin range. The referee quotes a value of 0.9 K. We do not know for which precise conditions the value has been estimated. However, these temperature values contradict the data which suggest the “effective” temperature is ≈ 500 mK, assuming the thermal hysteresis hypothesis holds. In our opinion, the correct interpretation must, thus, be that the simple models are insufficient, the numbers for the parameters are off, or we are missing some important pathway of heat dissipation. We do not haphazardly “discard” heating. Rather, our dynamical analysis of hysteresis yields excellent agreement with the experimental data. This indicates the “effective” temperature cannot be several hundred milli-Kelvin above base temperature of the refrigerator at the retrapping point in the I-V curve. (For higher “effective” temperatures, we expect the hysteresis in Figs. S1-3 to be smaller.) While we cannot provide a precise number without designing and running a new experiment that measures the temperature directly, we present strong evidence that the mechanism for retrapping is dynamical in nature---and the referee agrees as we understand. For this reason, we dismiss thermal hysteresis as the main cause of hysteresis in our experiment. We obviously did not make this connection clear enough in our original reply and hope the referee finds value in our line of reasoning.

I find the current way that Laroche is cited in the introduction to be unclear. The phrasing "A 4π -periodic Josephson effect in photon emission has also been reported for InAs nanowire junctions [17], where an on-chip detector was used. The seemingly conflicting theoretical and experimental results of recent years motivate our renewed interest in the topic, extending the scope of our analysis to nonlinear dynamics effects." casts doubt on Laroche. Yet, the discussion with the Referee says that the authors do not expect period-doubling dynamics for Laroche. I am not sure if this is intentional, but a straightforward reading would leave things unclear as to whether this result draws Laroche into question.

We thank the referee for pointing this out. Indeed, the wording can easily be misunderstood. To avoid a possible confusion, we changed it in the second revision of the manuscript to

“The seemingly conflicting aspects of earlier theoretical and experimental results, and the more recent experiments on other material systems motivate our renewed interest in the topic, extending the scope of our analysis to nonlinear dynamics effects.”

Reviewer #3 (Remarks to the Author):

Dear Editor,

The authors of the manuscript titled "Period-doubling in the phase dynamics of a shunted HgTe quantum well Josephson junction" have revised the manuscript to include a discussion of similar fractional Josephson experiments in nanowires (Ref 17). With the addition of this reference, the manuscript now includes a discussion of their results provide guidance for experiments in other systems showing fractional Josephson junction peaks. The manuscript presents a relatively complete understanding of a likely

mechanism for non-topological fractional Josephson peaks and how to avoid them. Therefore, I now believe that this manuscript is appropriate for the broad readership of Nat. Comm.

We hope we addressed all points of concern adequately.

List of additional changes in the second revision of the manuscript:

Main manuscript

Page 6, paragraph 2:

We removed footnote 38 of the previous version of the manuscript and reworded the sentence:

“Fig. 3a depicts a 2D map of the normalized microwave emission power at gate voltage $V_g = -0.08\text{V}$, when the Fermi level is close to the bottom of the conduction band subband, **and the critical current $I_c = 102\text{ nA}$. (We note that the position of the charge neutrality point shifts between cooldown cycles.)**”

Page 6, paragraph 2:

We removed footnote 40 and replaced it with reference 39 in the revised manuscript.

The figures and figure captions have been reworked to comply with the artwork guideline and the request that all abbreviations are described with the figure.

Figure 1:

“... (c) 2D map of the junction differential conductance G as a function of gate voltage V_g and magnetic field B ; conductance steps are labeled by the Landau level index ν ; ... (e) gate dependence of the normal state resistance R_N and critical current c as a function of gate voltage V_g **[or bulk carrier density n_e]; ...**”

Figure 2:

“... (a) Schematic of the RF measurement circuit [gate connection not shown]. **The bias current I_{DC} is generated by applying a voltage V_{DC} via a series resistor, followed by RC- and LC- low-pass filters (RC, LC). The voltage across the shunted sample V_S is amplified using an amplifier chain with a low-temperature (LTA), anchored at temperature $T = 2\text{ K}$, and two room-temperature amplifiers (RTA) with a total gain of $\sim 100\text{ dB}$. An attenuator (1 dB) provides thermal anchoring. The DC-averaged component \bar{V}_S is measured separately.** (b) photo of sample and shunt resistor mounted on and wired to the RF circuit board; (c) equivalent circuit used in numerical simulations.”

Figure 6:

“...**(b) DC-averaged junction current \bar{I}_J and power spectral density as function of \bar{V}_J at fixed $f_d = 5.2\text{ GHz}$**”

Supplementary information

Page 4, paragraph 3, line 3:

We corrected typos in the voltage values of the subgap features:

“It leads to features in the subgap resistance centered around the voltages $\sim 18 \mu\text{V}$ and $\sim 36 \mu\text{V}$.”

Page 16, Figure S12, panel b:

We corrected a typo in the units for power spectral density. The axis label reads now: “PSD_d ($\mu\text{W}/\text{Hz}$)”

On behalf of the authors,
With best regards,

Martin Stehno